# An Exploratory Study of the Enzymatic Hydroxycinnamoylation of Sucrose and Its Derivatives

**DOI:** 10.3390/molecules29174067

**Published:** 2024-08-28

**Authors:** Matej Cvečko, Vladimír Mastihuba, Mária Mastihubová

**Affiliations:** Institute of Chemistry, Slovak Academy of Sciences, 845 38 Bratislava, Slovakia; matej.cvecko@savba.sk (M.C.); vladimir.mastihuba@savba.sk (V.M.)

**Keywords:** phenylpropanoid sucrose esters, enzymatic hydroxycinnamoylation, regioselectivity, substrate specificity, sucrose ferulates and coumarates, Lipozyme TL IM, Pentopan 500 BG

## Abstract

Phenylpropanoid sucrose esters are a large and important group of natural substances with significant therapeutic potential. This work describes a pilot study of the enzymatic hydroxycinnamoylation of sucrose and its derivatives which was carried out with the aim of obtaining precursors of natural phenylpropanoid sucrose esters, e.g., vanicoside B. In addition to sucrose, some chemically prepared sucrose acetonides and substituted 3′-*O*-cinnamates were subjected to enzymatic transesterification with vinyl esters of coumaric, ferulic and 3,4,5-trimethoxycinnamic acid. Commercial enzyme preparations of Lipozyme TL IM lipase and Pentopan 500 BG exhibiting feruloyl esterase activity were tested as biocatalysts in these reactions. The substrate specificity of the used biocatalysts for the donor and acceptor as well as the regioselectivity of the reactions were evaluated and discussed. Surprisingly, Lipozyme TL IM catalyzed the cinnamoylation of sucrose derivatives more to the 1′-OH and 4′-OH positions than to the 6′-OH when the 3′-OH was free and the 6-OH was blocked by isopropylidene. In this case, Pentopan reacted comparably to 1′-OH and 6′-OH positions. If sucrose 3′-*O*-coumarate was used as an acceptor, in the case of feruloylation with Lipozyme in CH_3_CN, 6-*O*-ferulate was the main product (63%). Pentopan feruloylated sucrose 3′-*O*-coumarate comparably well at the 6-OH and 6′-OH positions (77%). When a proton-donor solvent was used, migration of the 3′-*O*-cinnamoyl group from fructose to the 2-OH position of glucose was observed. The enzyme hydroxycinnamoylations studied can shorten the targeted syntheses of various phenylpropanoid sucrose esters.

## 1. Introduction

Cinnamic acid sugar ester derivatives form a broad group of secondary plant metabolites with abundant structural diversity. They contain one or more cinnamoyl groups (Ph-CH=CH-CO-) or their derivatives that may be substituted with hydroxy- or methoxy-groups. They are linked to the non-anomeric carbon skeletons of the glycosyl group through ester bonds [1,2]. The best known and most studied are cinnamic acid sucrose esters, also called phenylpropanoid sucrose esters.

Phenylpropanoid sucrose esters (PPSEs) consist of a central sucrose (β-d-fructofuranosyl-α-d-glucopyranoside, **1**), which is variously decorated with one or more cinnamates (e.g., 3,4,5-trimethoxycinnamate, coumarate, ferulate, sinnapate, caffeate). Some sucrose hydroxyls may be acetylated or *p*-hydroxybenzoylated [3]. They are among the biologically active components isolated from phytomass and play several important roles in plant metabolism [4]. The variability of the structures of this type of compounds also brings about the variability of interesting biological activities useful for human health. Many of the isolated or synthesized cinnamoylated sucroses have antioxidant, photoprotective, anti-inflammatory, antidiabetic, antimicrobial, anticancer and antiviral effects [3,4]. There is mention that the position and number of phenolic and acetyl groups can influence the biological activity of PPSEs molecules. To establish an accurate structure-activity relationship with respect to α-glucosidase and α-amylase inhibition [5,6], even tailor-made synthetic, partially deprotected regioisomers were prepared [7]. 

Functionalization of unprotected or partially protected sucrose by base-catalyzed acylation usually affords a rich mixture of polysubstituted regioisomers. The distribution of the product also depends on the reactivity of the individual sucrose hydroxyls in the studied conditions. Regioselective acylations, especially of the secondary hydroxyls of sucrose, require protection, esterification and deprotection steps typical for classic sugar chemistry [8]. Acylation of the saccharides using phenolic acids also requires protection of the phenolic function and its deprotection in an appropriate phase of the synthesis. Judeh et al. prepared in this way several lapatosides and heloniosides from 2,1′:4,6-di-*O*-isopropylidene sucrose (**2**) [9,10]. Luan et al. synthesized tenuifoliside B by regioselective acylation to 3′-OH of sucrose by a cobalt chelate-directed reaction followed by Mitsunobu esterification of 6-OH [11]. Orthogonal protection/deprotection and selective cinnamoylation strategies were developed for the precise synthesis of PPSEs [12]. A similar principle was used in the past in the synthesis of galloylated sucrose [13].

Sucrose, as one of the most abundantly produced sugars in the world with a low price, is considered a multi-purpose organic raw material. Functional acylation of sucrose is the simplest way to obtain various surfactants, glycoconjugates or polymers [14,15]. The preparation of surfactants or amphiphilic molecules based on sucrose is one of the most studied fields in sucrose chemistry [16]. These compounds are now commonly prepared enzymatically using commercial lipases, since long fatty acids are their natural substrates [17].

Similarly, the problem of multistep synthesis of PPES could be avoided by introducing biocatalytic procedures that usually have a high degree of regioselectivity in the acylation of polyhydroxylated substrates. In addition, the phenolic groups of the activated hydroxycinnamate esters used as acyl donors do not need to be protected in such a process. The problem is that phenolic acids or their esters are not suitable substrates for most commercial lipases or proteases and pancreatic lipases are inhibited by them [18]. It should be mentioned that sucrose acetonides enzymatically acylated with dihydrocoumaric acid or its methyl ester were recently prepared [19]. This esterification using acid and transesterification using ester was catalyzed by CAL B (lipase from *Candida antarctica*). However, these conditions do not work when using hydroxycinnamic acids or their esters instead of the hydroxyarylalkyl analog. Karboune et al. used the feruloyl esterase expressed in Depol 740L from *Humicola* spp. and ferulic acid in a surfactant-free microemulsion medium for feruloylation of sucrose [20]. The yield of this reaction is reported (13%), but the regioselectivity is unknown. The structure of the product was proven only by APCI-MS analysis of the reaction mixture. In contrast to intensively studied surfactants based on sucrose esters, hydroxycinnamic sucrose esters have not yet been practically enzymatically prepared to our knowledge. The question remains as to which enzymes would be appropriate to use for these synthetic purposes.

Feruloyl esterases (FAEs) [E.C. 3.1.1.73] are defined as carboxylic acid esterases that catalyze the hydrolysis of the ester bond between hydroxycinnamic acids or their dehydrodimers and hemicellulolytic polysaccharides of plant cell walls [21,22,23,24]. Three classification systems have evolved over time. Crepin et al. in 2004 proposed to classify them into four types (A–D) based on their primary sequence homology and the similarity of hydrolytic esterase activity profiles towards methyl ester partially substituted hydroxycinnamic acids or their dimers [25]. In 2008 a new FAEs classification was proposed, based on the phylogenetic analysis of the available fungal genomes [26]. Finally, in 2016, a classification of 13 families based on 20 amino acid sequences of previously characterized FAEs and the alignment with 247 public fungal genomes was made [27]. 

Despite the fact that molecular techniques and genome sequencing have developed rapidly and the ABCD classification system no longer adequately represents the evolutionary relationships of fungal and bacterial FAEs, for our chemical reasoning this system works the best. On the basis of the classification in terms of their substrate specificity, type A FAEs show a preference for the phenolic moiety of the substrate that contains methoxy substitutions. These enzymes appear to prefer hydrophobic substrates with bulky substituents on the benzene ring. Their protein sequences are more similar to the lipase sequences. Type B FAEs prefer to hydrolyze substrates with hydroxy substituents and have sequence homology similar to acetyl xylan esterases. Finally, type C and D FAEs have broader substrate specificities with activities towards all model substrates and share sequence homology with chlorogenic acid esterases and tannases, while type D share sequence homology with xylanases and are the only group capable of hydrolyzing diferulates [25].

Unlike some lipases that work well in almost dry organic solvents, the catalyzed synthetic reactions of FAEs work best in ternary solvent systems forming microemulsions without detergents, since FAEs are often inactivated by exposure to organic solvents with a very low or no water content [28]. From an organic chemist’s point of view, water in the reaction mixture can sometimes be a problem if the acceptor contains water-sensitive functional groups or another acyl group that can migrate in the proton-donor solvents. In the case of using an activated acid ester as an acyl donor, this ester can spontaneously hydrolyze.

In the past, our group has been engaged in the search for commercial and readily available suitable biocatalysts capable of catalyzing the acylation of glycosides and saccharides with activated esters of phenolic acids [29,30]. We have prepared and tested several analytical substrates for the detection of suitable enzymes, as well as activated acyl donors required for synthesis [31,32,33,34]. This experience was used in the synthesis of several bioactive natural compounds [35,36]. 

*Thermomyces lanuginosus* (previously *Humicola lanuginosa*) fungal lipase [37] has feruloyl esterase activity and displays a high level of sequence identity with FAE from *Aspergillus niger* which was classified as FAE type A [38,39,40]. When a few mutations on a corresponding residue were constructed in a *Thermomyces lanuginosus* lipase, the enzyme obtained significant FAE activity. This enzyme is probably responsible for lipolytic activity of Lipolase 100T, Lipex 100T and Lipozyme TL IM, immobilized lipases supplied by Novozymes. All of these biocatalysts showed hydrolytic activity against 4-nitrophenyl esters of ferulic, caffeic and gallic acids. They also catalyzed the transesterification of vinyl esters of these acids with glycosides in organic solvents. In our previous report, we demonstrated the catalytic similarity of Lipolase 100T to FAEs type A through a synthetic profile in the transesterification of vinyl esters of various phenolic and non-phenolic aryl, arylalkyl and cinnamic acids with the model methyl-α-d-glucopyranoside. The hydrophobic character of aromatic donors was found to be very important for the FAE activity of Lipolase 100T [29]. The most effective enzyme preparation was Lipozyme TL IM. It worked reliably in the transesterification of vinyl and 2,2,2-trifluoroethyl gallate and caffeate [34,36]. The exceptional transferuloylation activity of Lipozyme was also tested in the preparation of various chromogenic substrates for feruloyl esterases by a French group [41].

Another enzyme preparation, Pentopan 500 BG, is a biocatalyst produced from the thermofilic fungus *Humicola insolens* that contains the hemicellulosic enzyme endo-xylanase and exhibits feruloyl esterase activities. Several works describe its ability to release free ferulic acid from maize bran [42] and catalysis of enantioselective acetylation of aromatic secondary alcohols [43,44]. Pentopan, in our previous study, had shown significant hydrolytic activity on *p*-nitrophenyl ferulate [29] and *p*-nitrophenyl gallate [34], and also activity in the transesterification of vinyl ferulate by glycosides [29].

Both immobilized enzyme preparations originating from thermophilic wood decay fungi are supplied by Novozymes and are very easily available and distributed, e.g., by Merck. Both enzymes were tested in hydroxycinnamoylations of free sucrose and its isopropylidenated and 3′-*O*-acylated derivatives in order to explore possible chemoenzymatic approaches to the preparation of tetra-*O*-hydroxycinnamoylated sucroses such as vanicoside B. This natural compound has shown remarkable anticancer [45,46] and antiviral [47] properties, and has been identified as a promising candidate for the treatment of Alzheimer’s disease [48]. 

## 2. Results and Discussion

### 2.1. Synthesis and General Remarks on the Selection of Acceptors and Donors

It has been shown that a simple order of reactivity of primary versus secondary alcohols does not always hold as a general rule for describing the relative reactivity of the eight hydroxyl groups of sucrose. Depending on the reaction conditions (solvent, electrophilic agent, type of catalysis), the nucleophilic behavior of sucrose changes, leading to compounds substituted in different positions [49]. In the case of enzyme catalysis, the modification of sucrose can lead to its different orientations in the active site of the enzyme.

Four acceptors, namely non-derivatized sucrose (**1**), 2,1′:4,6-di-*O*-isopropylidene sucrose (**2**) [50], 4,6-*O*-isopropylidene sucrose (**3**) [51], and 3′-*O*-cinnamoylated sucroses (**4a**–**b**), were prepared and used in the study. Compounds **4a**–**b** were prepared using the known complexation of sucrose with Co(II) [11,52]. Instead of anhydrides, more available acyl chlorides were used. The yields of 3′-*O*-coumarate **4a** and 3′-*O*-(3,4,5-trimethoxycinnamate) **4b** were rather low, but the reaction conditions were not optimized. Of the 3′-*O*-cinnamates, 3′-*O*-coumarate (**4a**) was chosen as the model, as the most frequently occurring phenylpropanoid sucrose in nature, and non-phenolic, also natural 3′-*O*-(3,4,5-trimethoxycoumarate) (**4b**) (Figure 1). Various activated cinnamoyl donors previously prepared [30], namely phenolic vinyl coumarate (**5a**) and vinyl ferulate (**5b**), as well as non-phenolic vinyl 3,4,5-trimethoxycinnamate (**5c**), were used for the acylation studies (Figure 1).

There were several reasons for believing that enzymatic hydroxycinnamoylation with partially protected or derivatized hydroxyls of sucrose would be a better solution. Sucrose is a disaccharide that has three primary hydroxyls and five secondary ones. Therefore, enzyme acylation, like chemical acylation, may not be sufficiently regioselective. Free sucrose is a natural substance and as such can be a substrate for secondary enzymes found in the rough enzyme preparation. In our case, glycosidases such as α-d-glucosidase or β-d-fructofuranosidase could split sucrose into basic sugars. After modification, sucrose would no longer be a suitable substrate for glycosidases. The last reason was practical. Free sucrose is not soluble in organic solvents in which the studied enzymes transesterificate well. Isopropylidenation or acylation will reduce the hydrophilic character of sucrose and increase its solubility in less polar solvents. Finally, in the case of partially protected or acylated sucroses, it is interesting to observe how the already existing group on the sucrose controls the subsequent enzyme acylation. Products of multiple acylation are welcome in this work because even natural phenylpropanoid sucroses are often decorated with several cinnamates.

### 2.2. Free Sucrose as an Acceptor in Enzymatic Feruloylation

The insolubility of free saccharides and decreased activity of hydrolases in organic solvents has provoked a search for convenient aprotic polar organic solvents, co-solvents or other unconventional conditions [17,53,54]. For our purposes, to achieve partial solubility of free sucrose and to maintain the activity of Lipozyme, 5% N,N-dimethylformamide in acetonitrile was found to be a suitable reaction medium. Approximately 11 mono- and more feruloylated derivatives of sucrose (**6a**–**d**), glucose (**7a**–**c**) and fructose (**8a**–**d**) were isolated from the reaction mixture and identified (Figure 1). This was also due to the fact that sucrose was decomposed into its monosaccharide components d-fructose and d-glucose during the reaction and various feruloylated α,β-glucopyranoses (9%) and α,β-fructoses (2%) were isolated (Figure 1).

Regarding the regioselectivity of feruloylation of these monosaccharides, in the case of fructose we learned that monoferuloylation did not proceed selectively. Among the six possible cyclic products, three products (**8b**–**d**, 1%) were identified in the reaction mixture. Since the fructose products were more difficult to identify from TLC and from the NMR spectra of the mixtures, it is possible that some fructose monoferulates were lost. Pure 1,6-di-*O*-feruloyl-α,β-d-fructofuranose **8a** (1%) feruloylated on primary hydroxyls was also isolated from fructose derivatives. In the case of d-glucose, 6-*O*-feruloyl-α,β-glucopyranoside **7c** (6%) and a mixture of 2,6- and 3,6-di-*O*-feruloyl-α,β-glucopyranosides **7a**–**b** (3%) were isolated. We also isolated various feruloylated sucroses in an approximately similar yield (11%). As can be seen in Figure 1, we obtained 6-*O*-feruloyl sucrose **6d** (5%), 6,1′-di-*O*-feruloyl sucrose **6c** (4%) and the tri-*O*-feruloylated products as a mixture (2%), namely 2,6,1′-tri-*O*-feruloyl sucrose **6a** and 6,1′,4′-tri-*O*-feruloyl sucrose **6b**.

The feruloylation of glucose and sucrose was more regioselective in the case of monoacylation. In both cases, only the primary 6-OH position was feruloylated. The product map for diferuloylated glucoses was more complicated. The **7b**α:**7b**β:**7a**α:**7a**β ratio was 3.6:2.0:2.4:1.0, which means that 3,6-diferulate slightly predominated over 2,6-diferulate. The existence of 6,1′-diferulated sucrose **6c** indicates that acylation to the 6′-OH position is less favorable. This also confirms the formation of surprising products of acylation at the secondary positions of 2-OH and 4′-OH **6a** and **6b** in a ratio of 1.9:1. No sucrose product had a 6′-OH position feruloylated. Lipozyme in acetonitrile can be proposed to catalyze the feruloylation of hydroxyl groups in free sucrose in the following order: 6-OH > 1′-OH > 2-OH ≥ 4′-OH. Comprehensive works mapping the enzymic acylations of sucrose by long fatty acids, for lipase from *Thermomyces lanuginosus* or *Humicola lanuginosa*, also report 6-OH acylation products as the main ones. Regarding the diacylation products, both 6,1′- and 6,6′-diacyls were found [16,17].

We consider this experiment to be positive, since the regioselectivity of feruloylation with Lipozyme on free fructose, glucose and sucrose was determined. The overall yield was not high, but this was also due to the fact that only two equivalents of the acyl donor were used and these were also consumed for the triferuloylated products. The occurrence of glucose and fructose can be explained by the enzymatic degradation of sucrose. According to our experience, different batches of Lipozyme and Pentopan comprise varying levels of invertase. In water, the enzymes do not hydrolyze methyl α-d-glucopyranoside, but partially hydrolyze sucrose. In the presence of ethanol, the enzymes may even form glycosides.

Using the same conditions, the reaction with Pentopan was also carried out. The reaction was stopped after three weeks, providing only traces of several products according to the TLC analysis. It is possible that Pentopan was deactivated by the presence of DMF in the reaction medium. For enzymes that normally hydrolyze amide or ester bonds to work in the esterification direction, the reactions have to be carried out in an organic medium with only the minimum amount of water necessary to maintain their active structure. However, the highly hydrophilic sucrose is insoluble in such conditions. Otherwise, with a higher concentration of water in the reaction, the activated donor—vinyl ester—is preferentially hydrolyzed by the hydrolase. Therefore, for acylation reactions with hydrolases requiring anhydrous and aprotic conditions, it is difficult to find a suitable solvent and temperatures that are compatible with the solubility of all substrates in the reaction and the stability and activity of the enzyme. On the other hand, this problem represents a challenge for the future in the field of studying less traditional reaction media.

### 2.3. Enzymatic Hydroxycinnamoylation of 2,1′:4,6-Di-O-isopropylidene Sucrose ***2***

Diisopropylidene sucrose **2** is a more rigid molecule, more soluble in less polar organic solvents. Sucrose derivative **2** as an acceptor has already been used in enzyme acylations in order to obtain 6′-*O*-monoesters. Reactions of **2** with myristic acid in toluene catalyzed by Lipozyme IM-60 (lipase from *M. miehei*), *Chromobacterium viscosum* lipase and Novozym 435 (lipase B from *Candida antarctica*) at 75 °C afforded at most 44% of the 6′-*O*-acylated products. The reaction time was from 2 to 10 days [55]. Novozym 435 also catalyzed the reaction of **2** with dihydrocoumaric acid at 60 °C. In 96 h of reaction, 27% of 6′-*O*-dihydrocoumarate was formed [19]. Despite the better solubility, cinnamoylation of acceptor **2** had the worst course with both of our biocatalysts. The reaction did not proceed even during a long reaction time of several weeks (Table 1, entries 1–4). A slow gradual hydrolysis of the vinyl esters of the substituted cinnamic acids **5a**–**c** was visible, probably due to the influence of atmospheric moisture penetrating into the reactions and at the same time the slow 2,1′-deisopropylidenation of **2**. Even, according to our experiences, relatively universal biocatalyst in hydroxycinnamoylations Lipozyme TL IM was not active, not even with lipophilic donor **5c**, friendly to lipases. It may be that the 6′-OH position of sucrose is really not favorable for Lipozyme TL IM. The usually less-active Pentopan proved to be a more effective biocatalyst in the hydroxycinnamoylation of **2** by hydroxycinnamates **5a**–**b** (Figure 2), while the reaction with ferulate took place with a higher yield (44%) in a shorter time (Table 1, entries 5 and 6). The reaction time was tens of days (Table 1). However, when trimethoxycinnamate **5c** was used, the reaction did not proceed as well. In this case, it turned out that Pentopan really has a reactivity closer to that of feruloyl esterases of type B and does not accept lipophilic donors. Product **9c** could not be prepared for understandable reasons (Table 1, entries 4 and 7).

### 2.4. Enzymatic Hydroxycinnamoylation of 4,6-O-isopropylidene Sucrose ***3***

The reaction of monoisopropylidene **3** with vinyl coumarate **5a** and vinyl ferulate **5b** under the catalysis of both selected enzymes was more successful in terms of yield but less regioselective (Figure 3). The reaction was studied in two solvents, CH_3_CN and sustainable 2-methyltetrahydrofuran (MeTHF).

In the case of acceptor **3** Lipozyme-catalyzed hydroxycinnamoylation (Table 2, entries 1,2,4,5), three monoacyls (1′-*O*-, 6′-*O*- and small amounts of 4′-*O*-) were obtained as products, as well as the corresponding diacyls (1′,6′-di-*O*- and 1′,4′-di-*O*-) (Figure 3). 1′-*O*-Monoacyls **10a**–**b** were the main products under the conditions used (around 30%). The yields of the individual coumaroylated products were in the following order: 1′-*O*- > 6′-*O*- > 1′,6′-di-*O*- ≥ 1′,4′-di-*O*- > 4′-*O*-. A small amount of triacylated products, including products acylated to position 2-*O*-(2,1′,6′- and 2,1′,4′-tri-*O*-coumarates), were isolated as a mixture from the reaction with coumarate **5a** (Table 2, entries 1, 2). The reaction with ferulate **5b** had a similar course in the case of Lipozyme and the same regioisomers were isolated. Diferulates **13b** and **14b** were obtained in lower yields compared to coumarates **13a** and **14a**. Tri-*O*-feruloylation products were observed only in traces (Table 2, entries 4, 5).

The reaction with Pentopan had a very slow course (tens of days) but was more regioselective. Products of acylation to the 4′-*O*-position were absent. Three products were formed in the order of 1′-*O*- ≅ 6′-*O*- > 1′,6′-di-*O*- (Table 2, entries 3, 6). Feruloylation catalyzed by Pentopan proceeded faster than coumaroylation and with higher yields in CH_3_CN, while no products were observed in MeTHF (Table 2, entries 3, 6 and 7).

The already mentioned work [55] reported 6′-*O*-myristate as the main product of enzyme acylation of 4,6-isopropylidenated sucrose **3** with myristic acid in the highest yield of 30%. It is not clear whether other byproducts were isolated. The authors of a recent paper [19] were unable to obtain any product when the Novozym 435 was used to catalyze dihydrocoumaroylation of acceptor **3** in *t*-butanol. 

### 2.5. Enzymatic Cinnamoylation of 3′-O-acylated Sucroses ***4a*** and ***4b***

To the best of our knowledge, 3′-*O*-acylated sucrose as starting material has not been subjected to enzymatic acylation so far. Reaction of sucroses **4a** and **4b** with cinnamoyl donors **5a**–**c** under catalysis of Lipozyme in CH_3_CN took place preferentially in the 6-OH position in our experiments. Products **15a**–**c** were isolated in yields of around 60%. (Figure 4 and Table 3, entries 1, 4, and 7). The minor products **17a**–**c** and **18a**–**c** were a mixture of products of diacylation to positions 6,6′-OH and 6,4′-OH. 6,3′,4′-Tri-*O*-acyls **17a**–**c** were sometimes formed in such small amounts that they could not be separated from 6,3′,6′-triacyl **18a**–**c** and their NMR characterization was determined on the basis of a comparison of the pure **18a**–**c** obtained from the reaction with Pentopan. For Lipozyme in CH_3_CN it can be said to catalyze the feruloylation of hydroxyl groups in 3′-*O*-acylated sucroses **4a**–**b** in the following order: 6-OH > 6′-OH > 4′-OH. In the reactions with Lipozyme in CH_3_CN, we did not observe the formation of acylation products to the 1′-OH position. Conversely, the 6′-OH position disadvantageous for acylation in acceptors **1**–**3** was involved in esterification in 3′-*O*-acylated sucroses **4a**–**b**. Probably, the conformation and reactivity of sucrose in the enzyme acylations studied are also influenced by the intramolecular hydrogen bond between 2-OH and 3′-OH in acceptors **1–3** and between 2-OH and 1′-OH in acceptors **4a**–**b**, because in this case position 3′-OH is substituted.

Lipozyme-catalyzed coumaroylation of **4a** was also carried out in MeTHF (entry 2). In this case, the reaction proceeded in the shortest time and the majority products were 6,3′,4′-(**17a**) and 6,3′,6′-(**18a**) tri-*O*-coumarates in a ratio of 1:3 and a total yield of 65%. In this case, repeated column chromatography succeeded in separating both regioisomers. Tetracoumarates (9%) were also isolated from acylation in MeTHF. After chromatography, the NMR analysis of the first fraction (2%) showed that three products were present in the sample. The two major products appeared to be 2,6,3′,4′- and product of migration 2,3,6,1′-tetracoumarates. The second fraction gave 6,1′,3′,6′-tetracoumarate **19a** as pure product (7%). This indicates that the reaction at higher degrees of acylation in MeTHF is already quite unselective, and minor acyl migration products are probably already present.

The reaction of sucrose **4b** with non-phenolic vinyl ester **5c**, catalyzed by Lipozyme, was also carried out in *t*-amyl alcohol. However, in this proton donor solvent, the product of acyl migration from the 3′-*O*-position of fructose to the 2-*O*-position of glucose **20** (5%) was also observed (Table 3, entry 8).

Hydroxycinnamoylation of **4a**–**b** catalyzed by Pentopan in CH_3_CN proceeds approximately equally in the 6-OH ≅ 6′-OH positions (Table 3, entries 3 and 5). The reaction time was again in tens of days. Although the reaction was slow, the enzyme was active all that time in CH_3_CN. Hydrolysis of vinyl esters was minimal. In the same time, the reaction with vinyl ferulate compared to vinyl coumarate gave almost twice the yield of monoferulates (**15b** and **16b**) and almost three times the yield of diferulate **18b** (Table 3, entry 5). We believe that in the case when more enzyme and more equivalents of acyls are used under prolonged reaction time, it is possible to obtain a regioselective 6,3′,6′-triacyl derivatives **18a** and **18b** as the majority product. Again, Pentopan was not active in MeTHF and no products were observed (Table 3, entry 6). No products were observed even when Pentopan-mediated acylation with vinyl trimethoxycinnamate was attempted (Table 3, entry 9). This again confirms that Pentopan has a narrower substrate specificity towards acyl donors. It would also be appropriate to investigate its reactivity in another organic solvents, as it is different from that of lipases.

## 3. Materials and Methods

### 3.1. General

The reactions were performed with commercial reagents purchased from Merck (Darmstadt, Germany), Sigma-Aldrich (St. Louis, MO, USA), Acrōs Organics (Geel, Belgium). Lipozyme^®^ TL IM and Pentopan^®^ 500 BG were from Novozymes (Bagsværd, Denmark). Molecular sieves (4 Å) were microwaved before use. TLC was performed on aluminum sheets precoated with silica gel 60 F_254_ (Merck, Darmstadt, Germany). Spots were visualized by UV light (λ_max_ = 254 nm) and charred with 5% ethanolic sulfuric acid comprising 1% orcinol. The solvents used in the enzymatic reactions, *t*-amyl alcohol (*t*-AmOH), acetonitrile (CH_3_CN) and 2-methyltetrahydrofuran (MeTHF), were of HPLC grade and pre-dried over molecular sieves. Column chromatography was carried out on silica gel SiliaFlash P60/40–63 μm, 60 Å (SiliCycle Inc., Québec, QC, Canada) using distilled solvents (ethyl acetate (EA), chloroform or methanol (MeOH)). 

^1^H NMR and ^13^C NMR spectra were recorded at 25 °C on 400 MHz Bruker AVANCE III HD equipped with a Prodigy CryoProbe (Bruker GmbH, Karlsruhe, Germany). Copies of ^1^H and ^13^C NMR spectra for selected cinnamoylated sucroses are attached as Appendix A. Chemical shifts were referenced to CD_3_OD (δ 3.31 ppm for ^1^H, δ 49.0 for ^13^C) and to CD_3_COCD_3_ (δ 2.05 ppm for ^1^H, δ 29.8 for ^13^C). The acetone-d_6_ was used with 10% D_2_O to exchange OH for OD. Chemical shifts (in ppm) and coupling constants (in Hz) were obtained by first-order analysis; assignments were derived from two-dimensional homonuclear (H-H COSY) and heteronuclear (HSQC) spectra. Optical rotations were measured on a Jasco P2000 (Jasco Products Company, Oklahoma City, OK, USA) polarimeter at 20 °C. Ion source HESI (heated electrospray) was performed on the mass analyzer orbitrap (Orbitrap Elite Thermo Scientific, Thermo Fisher Scientific, MA, USA), with the following settings: capillary temperature 350 ºC, source heater temperature 300 °C, mass range 80–900 *m/z*, full scan, positive polarity, resolution 120,000.

### 3.2. Feruloylation of Free Sucrose Using Lipozyme TL IM

Sucrose (**1**) (1.2 g, 0.0035 mol), vinyl ferulate (1.54 g, 0.007 mol, 2 equiv.) and Lipozyme TL IM (2.5 g) were shaken in dry CH_3_CN:DMF (20 mL, 19:1) for 96 h at 37 °C. The mixture was then filtered through Celite 545 and the filter cake was washed several times with hot acetone. The combined filtrates were concentrated and the final mixture was purified by silica gel column chromatography (CHCl_3_:MeOH 19:1, 14:1, 9:1, 4:1). We gradually obtained the following products from preparative chromatography 1′,6-di-*O*-feruloyl-α,β-d-fructofuranose (**8a**) (0.017 g, 1%); a mixture of diferuloylated glucoses: 2,6-di-*O*-feruloyl-α,β-d-glucopyranose (**7a**) and 3,6-di-*O*-feruloyl-α,β-d-glucopyranose (**7b**) (0.055 g, 3%); a mixture of triferuloylated sucroses: 2,6,1′-tri-*O*-feruloyl sucrose (**6a**) and 6,1′,4′-tri-*O*-feruloyl sucrose (**6b**) (0.049 g, 2%); 6,1′-di-*O*-feruloyl sucrose (**6c**) (0.089 g, 4%); a mixture of three monoferuloylated fructoses 6-*O*-feruloyl-β-d-fructofuranose (**8b**), 1-feruloyl-α-d-fructofuranose (**8c**) and 6-*O*-feruloyl-β-d-fructopyranose (**8d**) (0.015 g, 1%); 6-*O*-feruloyl-α,β-glucopyranose (**7c**) (0.079, 6%) and finally, 6-*O*-feruloyl sucrose (**6d**) (0.093 g, 5%) as major product in addition to another product that has not been identified (probably 1′-*O*-regioisomer). 

NMR signals of products in the mixture: 1′,6-Di-*O*-feruloyl-α,β-d-fructofuranose (**8a**, α:β—1:2.4), R_f_ = 0.65 (CHCl_3_:MeOH—3:1). ^1^H NMR (400 MHz, CD_3_OD) δ 7.66 (2×d, *J* = 15.8 Hz, 2×H-Aβ), 7.63 (d, *J* = 15.9 Hz, H-Aα), 7.61 (d, *J* = 15.9 Hz, H-Aα), 7.19 (d, *J* = 1.9 Hz, H-Phβ), 7.17 (d, *J* = 2.0 Hz, H-Phβ), 7.12 (d, *J* = 2.0 Hz, H-Phα), 7.08 (dd, *J* = 8.2, 2.0 Hz, H-Phβ), 7.06 (dd, *J* = 8.2, 2.1 Hz, H-Phβ), 7.03 (dd, *J* = 8.3, 2.0 Hz, H-Phα), 6.99 (dd, *J* = 8.2, 2.0 Hz, H-Phα), 6.81 (d, *J* = 8.2 Hz, H-Phβ), 6.80 (d, *J* = 8.3 Hz, H-Phβ), 6.78 (d, *J* = 8.1 Hz, H-Phα), 6.76 (d, *J* = 8.1 Hz, H-Phα), 6.41 (d, *J* = 15.9 Hz, H-Bβ), 6.38 (d, *J* = 15.9 Hz, H-Bβ), 6.38 (d, *J* = 16.0 Hz, H-Bα), 6.33 (d, *J* = 15.9 Hz, H-Bα), 4.44 (dd, *J* = 11.9, 3.4 Hz, H-6aα), 4.41 (dd, *J* = 11.8, 3.5 Hz, H-6aβ), 4.28 (dd, *J* = 11.8, 6.7 Hz, H-6bα), 4.28 (d, *J* = 11.7 Hz, H-1aα), 4.27 (dd, *J* = 11.7, 6.8, H-6bβ), 4.26 (d, *J* = 11.6 Hz, H-1aβ), 4.22–4.16 (m, H-5α), 4.15 (t, *J* = 7.4 Hz, H-4β), 4.16 (d, *J* = 11.6 Hz, H-1bβ), 4.10 (d, *J* = 5.7 Hz, H-3α), 4.08 (d, *J* = 7.7 Hz, H-3β), 4.03 (dd, *J* = 7.0, 5.6 H-4α), 3.99 (td, *J* = 6.9, 3.4 Hz, H-5β), 3.88 (2×s, OCH_3_β), 3.86 (s, OCH_3_α), 3.85 (s, OCH_3_α). ^13^C NMR (101 MHz, CD_3_OD) δ 169.1 (COOβ), 169.0 (COOα), 168.9 (COOα), 168.6 (COOβ), 150.7 (C-Ph), 150.6 (C-Ph), 150.6 (C-Ph), 149.4 (C-Phβ), 149.3 (C-Phβ), 149.3 (C-Phα), 149.3 (C-Phα), 147.3 (CH-Aβ), 147.2 (CH-Aα), 147.1 (CH-Aβ), 147.1 (CH-Aα), 127.7 (C-Phβ), 127.7 (C-Phα), 127.7 (C-Phβ), 127.6 (C-Phα), 124.2 (CH-Phβ),, 124.1 (CH-Ph(α,β)), 124.1 (CH-Phα), 116.5 (CH-Phβ), 116.4 (CH-Phα), 115.4 (CH-Bα), 115.3 (CH-Bβ), 115.1 (CH-B(α,β)), 111.8 (CH-Phα), 111.7 (CH-Phβ), 105.0 (C-2α), 101. 9 (C-2β), 84.0 (C-3α), 80.6 (C-5α)), 80.4 (C-5β), 78.3 (C-4α), 78.2 (C-3β), 77.2 (C-4β), 66.6 (C-6β), 66.1, 65.7, 64.9 (C-6α), 56.5 (2×OCH_3_ (β)), 56.4 (2×OCH_3_ (α)).
3,6-Di-*O*-feruloyl-α-d-glucopyranose (**7b**α); 2,6-di-*O*-feruloyl-α-d-glucopyranose (**7a**α); 3,6-di-*O*-feruloyl-β-d-glucopyranose (**7b**β); 2,6-di-*O*-feruloyl-β-d-glucopyranose (**7a**β) (3.6:2.4:2:1), R_f_ = 0.63 (CHCl_3_:MeOH–3:1). ^1^H NMR (400 MHz, CD_3_OD) δ 7.71 (d, *J* = 15.9 Hz, H-A), 7.68 (d, *J* = 15.9 Hz, H-A), 7.66 (d, *J* = 16.0 Hz, H-A), 7.65 (d, *J* = 16.0 Hz, H-A), 7.23, 7.21, 7.20, 7.17 (4xbs, H-Ph), 7.10, 7.09, 7.09 (4xd, *J* = 8.3 Hz, H-Ph), 6.83 (bdd, *J* = 8.2, 1.8 Hz, H-Ph), 6.46 (d, *J* = 16.1 Hz, H-B), 6.44 (d, *J* = 15.9 Hz, H-B), 6.42 (d, *J* = 16.1 Hz, H-B), 6.41 (d, *J* = 15.9 Hz, H-B), 5.38 (t, *J* = 9.4 Hz, H-3 (**7b***α*)), 5.37 (d, *J* = 4.0 Hz, H-1 (**7a***α*)), 5.20 (d, *J* = 3.7 Hz, H-1 (**7b***α*)), 5.11 (t, *J* = 9.3 Hz, H-3 (**7b***β*)), 4.85 (t overlapped with HDO, H-2 (**7a***β*), 4.77 (d, *J* = 7.9 Hz, H-1 (**7a***β*)), 4.74 (dd, J = 10.0, 3.7 Hz, H-2 (**7a***α*)), 4.67 (d, *J* = 7.8 Hz, H-1 (**7b***β*)), 4.60–4.46 (m, 4×H-6a), 4.44–4.34 (m, 4×H-6b), 4.19 (ddd, *J* = 10.3, 5.1, 2.2 Hz, H-5 (**7b***α*)), 4.15–4.10 (m, H-5, (**7a***α*)), 4.01 (dd, *J* = 10.0, 8.9 Hz, H-3 (**7a***α*)), 3.91, 3.90 (8xs, OCH_3_), 3.71 (ddd, *J* = 7.6, 4.6, 1.7 Hz, H-5, (**7b***β*)), 3.68–3.63 (m, H-5, (**7a***β*)), 3.66 (t, *J* = 9.2, H-3 (**7a***β*), 3.65 (dd, *J* = 9.8, 3.7 Hz, H-2 (**7b***α*)), 3.64 (t, *J* = 9.6 Hz, H-4 (**7b***β*)), 3.63 (t, *J* = 9.7 Hz, H-4 (**7b***α*)), 3.53 (t, *J* = 9.6 Hz, H-4 (**7a***α*)), 3.52 (t, *J* = 9.5 Hz, H-4 (**7a***β*)), 3.41 (dd, *J* = 9.6, 7.8 Hz, H-2 (**7b***β*)). ^13^C NMR (101 MHz, CD_3_OD) δ 169.3, 2×169.1, 3×169.0, 168.8, 168.5 (8×COO), 98.3 (C-1 (**7b***β*)), 96.6 (C-1 (**7a***β*)), 94.0 (C-1 (**7b***α*)), 91.4 (C-1 (**7a***α*)).2,6,1′-Tri-*O*-feruloyl sucrose (**6a**) and 6,1′,4′-tri-*O*-feruloyl sucrose (**6b**) (1.9:1), R_f_ = 0.47 (CHCl_3_:MeOH—3:1). ^1^H NMR (400 MHz, CD_3_OD) δ 7.70–7.59 (m, 5×H-A), 7.50 (d, *J* = 15.9 Hz, H-A (**6a**)), 7.31–6.98 (m, 12H-Ph), 6.81 (d, *J* = 8.2 Hz, H-Ph (**6a**)), 6.81 (d, *J* = 8.2 Hz, H-Ph (**6b**)), 6.80 (d, *J* = 8.2 Hz, H-Ph (**6b**)), 6.79 (d, *J* = 8.2 Hz, H-Ph (**6a**)), 6.77 (d, *J* = 8.2 Hz, H-Ph (**6a**)), 6.75 (d, *J* = 8.2 Hz, H-Ph (**6b**)), 6.54 (d, *J* = 16.0 Hz, H-B (**6b**)), 6.41 (d, *J* = 15.9 Hz, H-B (**6b**)), 6.38 (d, *J* = 15.9 Hz, H-B (**6b**)), 6.47 (d, *J* = 15.9 Hz, (**6a**)), 6.40 (d, *J* = 15.9 Hz, (**6a**)), 6.32 (d, *J* = 15.9 Hz, H-B (**6a**)), 5.74 (d, *J* = 3.7 Hz, H-1 (**6a**)), 5.53 (d, *J* = 3.8 Hz, H-1 (**6b**)), 5.41 (t, *J* = 7.5 Hz, H-4′ (**6b**)), 4.84 (dd, H-2 (**6a**) overlapped with HDO), 4.61 (bd, *J* = 10.0 Hz, H-6a (**6b**)), 4.56 (dd, *J* = 11.5, 1.7 Hz, H-6a (**6a**)), 4.51 (d, *J* = 12.2 Hz, H-1′a (**6b**)), 4.49 (d, *J* = 7.9 Hz, H-3′ (**6b**)), 4.35 (d, *J* = 12.2 Hz, H-1′b (**6b**)), 4.32 (dd, *J* = 11.7, 6.5 Hz, H-6b (**6a**)), 4.26 (d, *J* = 11.9 Hz, H-1′a (**6a**)), 4.29–4.21 (m, H-5 (**6a**), H-5 (**6b**) and H-6b (**6b**)), 4.17 (d, *J* = 11.9 Hz, H-1′b (**6a**)), 4.14 (d, *J* = 8.5 Hz, H-3′ (**6a**)), 4.10 (t, *J* = 8.2 Hz, H-4′ (**6a**)), 4.09–3.98 (m, H-5′ (**6b**)), 4.05 (t, *J* = 9.6 Hz, H-3 (**6a**)), 4.01–3.76 (m, H-6′a (**6b**), H-6′a (**6a**), H-6′b (**6b**), H-5′ (**6a**) and H-6′b (**6a**)), 3.89, 3.89, 3.88, 3.88, 3.85, 3.83 (6xs, OCH_3_), 3.77 (t, *J* = 9.3 Hz, H-3 (**6b**)), 3.76 (dd, H-6′ (**6a**)), 3.50 (dd, *J* = 9.8, 3.8 Hz, H-2 (**6b**)), 3.48 (t, *J* = 9.5 Hz, H-4 (**6a**)), 3.32 (t, H-4 (**6b**) overlapped with CD_3_OD). ^13^C NMR (101 MHz, CD_3_OD) δ 169.4, 169.2, 168.9, 168.5, 2×168.4 (6×COO), 104.7 (C-2′ (**6b**)), 103.9 (C-2′ (**6a**)), 94.0 (CH-1 (**6b**)), 91.1 (CH-1 (**6a**)).6,1′-Di-*O*-feruloyl sucrose (**6c**). White amorphous solid; ${\left[\alpha \right]}_{D}^{20}$ = +11.1° (*c* = 1.0, CH_3_OH), R_f_ = 0.50 (CHCl_3_:MeOH—3:1). ^1^H NMR (400 MHz, CD_3_OD) δ 7.63 (2×d, *J* = 15.9 Hz, 2H, 2×H-A), 7.21 (d, *J* = 1.9 Hz, 1H, H-Ar), 7.18 (d, *J* = 2.0 Hz, 1H, H-Ar), 7.08 (dd, *J* = 8.2, 1.9 Hz, 1H, H-Ar), 7.07 (dd, *J* = 8.2, 1.9 Hz, 1H, H-Ar), 6.81 (d, *J* = 8.2 Hz, 1H, H-Ar), 6.80 (d, *J* = 8.1 Hz, 1H, H-Ar), 6.43 (d, *J* = 15.9 Hz, 1H, H-B), 6.37 (d, *J* = 15.9 Hz, 1H, H-B), 5.47 (d, *J* = 3.8 Hz, 1H, H-1), 4.53 (dd, *J* = 11.9, 2.0 Hz, 1H, H-6a), 4.44 (d, *J* = 12.1 Hz, 1H, H-1′a), 4.29 (d, *J* = 12.1 Hz, 1H, H-1′b), 4.27 (dd, *J* = 11.9, 6.4 Hz, 1H, H-6b), 4.19–4.13 (m, 1H, H-5), 4.18 (d, *J* = 8.5 Hz, 1H, H-3′), 4.11 (t, *J* = 8.1 Hz, 1H, H-4′), 3.89 (s, 3H, OCH_3_), 3.88 (s, 3H, OCH_3_), 3.87–3.77 (m, 3H, H-6′a, H-5′, H-6′b), 3.74 (dd, *J* = 9.8, 8.9 Hz, 1H, H-3), 3.48 (dd, *J* = 9.8, 3.9 Hz, 1H, H-2), 3.34 (t, *J* = 9.6 Hz, 1H, H-4). ^13^C NMR (101 MHz, CD_3_OD) δ 169.2 (COO), 168.5 (COO), 150.7 (C-Ar), 150.6 (C-Ar), 149.4 (C-Ar), 149.3 (C-Ar), 147.4 (CH-A), 147.1 (CH-A), 127.7 (C-Ar), 127.6 (C-Ar), 124.3 (CH-Ar), 124.3 (CH-Ar), 116.5 (CH-Ar), 116.4 (CH-Ar), 115.3 (CH-B), 115.0 (CH-B), 111.7 (2×CH-Ar), 104.3 (C-2′), 93.9 (C-1), 84.0 (C-5′), 78.8 (C-3′), 75.6 (C-4′), 74.6 (C-3), 73.0 (C-2), 72.1 (C-5), 72.0 (C-4), 65.1 (C-6), 64.1 (C-1′), 64.0 (C-6′), 56.5 (2×OCH_3_). HRMS (ESI): *m/z* calcd. for C_32_H_38_O_17_Na ([M + Na]^+^) 717.20012, found 717.20015.1-Feruloyl-β-d-fructopyranose (**8d**), 1-feruloyl-α-d-fructofuranose (**8b**), 6-feruloyl-β-d-fructofuranose (**8c**). (1:1.2:1.4), R_f_ = 0.42 (CHCl_3_:MeOH–3:1). ^13^C NMR (101 MHz, CD_3_OD) δ 105.2 (C-2, **8b**), 103.2 (C-2, **8c**), 98.2 (C-2, **8d**).6-O-Feruloyl-α,β-d-glucopyranose. (**7c**, α:β—1:1), R_f_ = 0.39 (CHCl_3_:MeOH—3:1). ^1^H NMR (400 MHz, CD_3_OD) δ 7.63 (2×d, *J* = 15.9 Hz, 2H, 2×H-A), 7.18 (2×d, *J* = 1.9 Hz, 2H, 2×H-Ph), 7.07 (2×dd, *J* = 8.2, 2.0 Hz, 2H, 2×H-Ph), 6.82 (2×d, *J* = 8.1 Hz, 2H, 2×H), 6.36 (2×d, *J* = 15.9 Hz, 2H, 2×H-B), 5.11 (d, *J* = 3.7 Hz, 1H, H-1α), 4.51 (d, *J* = 7.9 Hz, 1H, H-1β), 4.51 (dd, *J* = 11.9, 2.2 Hz, 1H, H-6a), 4.46 (dd, *J* = 11.9, 2.3 Hz, 1H, H-6a), 4.34 (dd, *J* = 11.9, 5.5 Hz, 2H, H-6b), 4.31 (dd, *J* = 12.0, 5.8 Hz, 1H, H-6b), 4.04 (ddd, *J* = 10.1, 5.4, 2.3 Hz, 1H, H-5), 3.89 (2×s, 6H, 2×OCH_3_), 3.70 (t, *J* = 9.2 Hz, 1H, H-3α), 3.55 (ddd, *J* = 9.7, 5.8, 2.1 Hz, 1H, H-5), 3.42–3.33 (m, 4H, 2×H-4, H-2α, H-3β), 3.17 (dd, *J* = 9.2, 7.9 Hz, 1H, H-2β). ^13^C NMR (101 MHz, CD_3_OD) δ 169.2 (COO), 169.1 (COO), 150.8 (C-Ph), 150.7 (C-Ph), 149.5 (2×C-Ph), 147.0 (CH-A), 146.9 (CH-A), 127.8 (2×C-Ph), 124.1 (2×CH-Ph), 116.6 (2×CH-Ph), 115.4 (2×CH-B), 111.9 (2×CH-Ph), 98.3 (C-1β), 94.1 (C-1α), 76.3 (C-2β), 75.6 (C-5), 74.9 (C-3α), 78.0, 73.9, 72.1, 71.8 (C-2α, C-3β, 2×C-4), 70.9 (C-5), 65.0, 64.9 (2×C-6), 56.6 (2×OCH_3_).6-*O*-Feruloyl sucrose (**6d**), R_f_ = 0.25 (CHCl_3_:MeOH—3:1). ^1^H NMR (400 MHz, CD_3_OD) δ 7.65 (d, *J* = 15.9 Hz, 1H, H-A), 7.19 (d, *J* = 2.0 Hz, 1H, H-Ar), 7.08 (dd, *J* = 8.3, 2.0 Hz, 1H, H-Ar), 6.80 (d, *J* = 8.1 Hz, 1H, H-Ar), 6.37 (d, *J* = 15.9 Hz, 1H, H-B), 5.40 (d, *J* = 3.8 Hz, 1H, H-1), 4.50 (dd, *J* = 11.7, 7.4 Hz, 1H, H-6a), 4.42 (dd, *J* = 11., 3.3 Hz, 1H, H-6b), 4.13 (d, *J* = 8.2 Hz, 1H, H-3′), 4.08 (t, *J* = 7.9 Hz, 1H, H-4′), 3.99 (td, *J* = 11.5, 3.2 Hz, 1H, H-5), 3.90 (s, 3H, OCH_3_), 3.89–3.81 (m, 2H, H-6′a, H-5′), 3.77–3.69 (m, 2H, H-6′b, H-3), 3.67 (d, *J* = 12.3 Hz, 1H, H-1′a), 3.63 (d, *J* = 12.5 Hz, 1H, H-1′b), 3.48 (dd, *J* = 9.8, 3.8 Hz, 1H, H-2), 3.34 (t, *J* = 9.5 Hz, 1H, H-4). ^13^C NMR (101 MHz, CD_3_OD) δ 105.6 (C-2′), 93.5 (CH-1).

### 3.3. General Procedure for the Enzymatic Hydroxycinnamoylation of 2,1′:4,6-Di-O-isopropylidene Sucrose ***2***

Diisopropylidene sucrose **2 [50]** (440 mg, 1 mmol), vinyl hydroxycinnamate **5a** or **5b** (2 mmol, 2 equiv.) and enzyme (Lipozyme TL IM (1 g) or Pentopan (1 g)) were suspended in dry CH_3_CN (20 mL) and shaken at 37 °C for several days (Table 1). The reaction was monitored by TLC and stopped by filtration through Celite 545. The filter cake was washed several times with hot ethyl acetate. The filtrate was combined with washings, concentrated and the product was purified by column chromatography (ethyl acetate). The yields of products are presented in Table 1.
6′-*O*-Coumaroyl-2,1′:4,6-di-*O*-isopropylidene sucrose (**9a**). White amorphous solid; ${\left[\alpha \right]}_{D}^{20}$ = +55.9° (*c* = 1.0, CH_3_OH). ^1^H NMR (400 MHz, CD_3_OD) δ 7.64 (d, *J* = 15.9 Hz, 1H, H-A), 7.46 (d, *J* = 8.7 Hz, 2H, H-Ph), 6.81 (d, *J* = 8.6 Hz, 2H, H-Ph), 6.35 (d, *J* = 15.9 Hz, 1H, H-B), 6.05 (d, *J* = 3.7 Hz, 1H, H-1), 4.47–4.41 (m, 1H, H-6′a), 4.31–4.22 (m, 1H, H-6′b), 4.13 (d, *J* = 12.3 Hz, 1H, H-1′a), 4.08–4.03 (m, 2H, H-4′, H-5′), 3.92–3.82 (m, 3H, H-3, H-5, H-6a), 3.75–3.67 (m, 3H, H-2, H-6b, H-3′), 3.55 (t, *J* = 9.4 Hz, 1H, H-4), 3.44 (d, *J* = 12.3 Hz, 1H, H-1′b), 1.50 (s, 3H, CH_3_), 1.47 (s, 3H, CH_3_), 1.38 (s, 6H, 2×CH_3_). ^13^C NMR (101 MHz, CD_3_OD) δ 169.0 (COO), 161.2 (C-Ph),146.7 (CH-A), 131.2 (CH-Ph), 127.2 (C-Ph), 116.8 (CH-Ph), 115.0 (CH-B), 105.2 (C-2′), 102.8 (C-izopr), 100.8 (C-izopr), 92.4 (C-1), 81.1 (C-5′), 80.0 (C-3′), 77.9 (C-4′), 75.2 (C-2), 74.8 (C-4), 70.9 (C-3), 67.5 (C-6′), 66.9 (C-1′), 64.6 (C-5), 63.4 (C-6), 29.5 (CH_3_), 25.5 (CH_3_), 24.4 (CH_3_), 19.3 (CH_3_). HRMS (ESI): *m/z* calcd. for C_27_H_36_O_13_Na ([M + Na]^+^) 591.20481, found 591.20485.6′-*O*-Feruloyl-2,1′:4,6-di-*O*-isopropylidene sucrose (**9b**). White amorphous solid; ${\left[\alpha \right]}_{D}^{20}$ = +57.7° (*c* = 1.0, CH_3_OH). ^1^H NMR (400 MHz, CD_3_OD) δ 7.63 (d, *J* = 15.9 Hz, 1H, H-A), 7.07 (dd, *J* = 8.2, 2.0 Hz, 1H, H-Ph), 6.81 (d, *J* = 8.2 Hz, 2H, H-Ph), 6.38 (d, *J* = 15.9 Hz, 1H, H-B), 6.05 (d, *J* = 3.8 Hz, 1H, H-1), 4.48–4.42 (m, 1H, H-6′a), 4.30–4.22 (m, 1H, H-6′b), 4.13 (d, *J* = 12.3 Hz, 1H, H-1′a), 4.08–4.03 (m, 2H, H-4′, H-5′), 3.89 (s, 3H, OCH_3_), 3.89–3.83 (m, 3H, H-3, H-5, H-6a), 3.75–3.66 (m, 3H, H-2, H-6b, H-3′), 3.55 (t, *J* = 9.3 Hz, 1H, H-4), 3.44 (d, *J* = 12.3 Hz, 1H, H-1′b), 1.50 (s, 3H, CH_3_), 1.47 (s, 3H, CH_3_), 1.38 (s, 6H, 2×CH_3_). ^13^C NMR (101 MHz, CD_3_OD) δ 168.9 (COO), 150.6 (C-Ph), 149.3 (C-Ph), 147.0 (CH-A), 127.7 (C-Ph), 124.1 (CH-Ph), 116.5 (CH-Ph), 115.3 (CH-B), 111.7 (CH-Ph), 105.2 (C-2′), 102.8 (C-izopr), 100.8 (C-izopr), 92.4 (C-1), 81.1 (C-5′), 80.0 (C-3′), 77.9 (C-4′), 75.2 (C-2), 74.8 (C-4), 70.9 (C-3), 67.5 (C-6′), 66.9 (C-1′), 64.6 (C-5), 63.4 (C-6), 56.5 (OCH_3_), 29.5 (CH_3_), 25.5 (CH_3_), 24.4 (CH_3_), 19.4 (CH_3_). HRMS (ESI): *m/z* calcd. for C_28_H_38_O_14_Na ([M + Na]^+^) 621.21538; found 621.21587.

### 3.4. General Procedure for the Enzymatic Hydroxycinnamoylation of 4,6-O-isopropylidene Sucrose ***3***

Monoisopropylidene sucrose **3 [51]** (0.382 mg, 1 mmol), molecular sieves 4Å (1 g), vinyl hydroxycinnamate **5a** or **5b** (3 mmol, 3 equiv.), enzyme (Lipozyme TL IM (1 g) or Pentopan (1 g)) were suspended in dry CH_3_CN (20 mL) and shaken at 37 °C for several hours (Table 2). The reaction was monitored by TLC and stopped by filtration through Celite 545. The filter cake was washed several times with hot ethyl acetate. The filtrate was combined with washings, concentrated and the product was purified by column chromatography (ethyl acetate → ethyl acetate:methanol—9:1). The yields of individual regioisomers are presented in Table 2.
1′-*O*-Coumaroyl-4,6-*O*-isopropylidene sucrose (**10a**). White amorphous solid, ${\left[\alpha \right]}_{D}^{20}$ = +15.2° (*c* 1.0, CH_3_OH). ^1^H NMR (400 MHz, CD_3_OD) δ 7.65 (d, *J* = 15.9 Hz, 1H, H-A), 7.47 (d, *J* = 8.7 Hz, 2H, H-Ph), 6.81 (d, *J* = 8.7 Hz, 2H, H-Ph), 6.36 (d, *J* = 15.9 Hz, 1H, H-B), 5.43 (d, *J* = 4.0 Hz, 1H, H-1), 4.42 (d, *J* = 12.1 Hz, 1H, H-1′a), 4.23 (d, *J* = 12.1 Hz, 1H, H-1′b), 4.16 (d, *J* = 8.6 Hz, 1H, H-3′), 4.09 (t, *J* = 8.4 Hz, 1H, H-4′), 3.87 (t, *J* = 9.7 Hz, 1H), 3.94–3.83 (m, 2H, H-5, H-6a), 3.82–3.66 (m, 5H, H-5′, H-3, H-6′a, H-6′b, H-6b) 3.51 (dd, *J* = 9.5, 3.8 Hz, 1H, H-2), 3.51 (t, *J* = 9.5 Hz, 1H, H-4), 1.50 (s, 3H, CH_3_), 1.38 (s, 3H, CH_3_). ^13^C NMR (101 MHz, CD_3_OD) δ 168.4 (COO), 161.4 (C-Ph), 147.1 (CH-A), 131.3 (CH-Ph), 127.1 (C-Ph), 116.8 (CH-Ph), 114.7 (CH-B), 104.2 (C-2′), 100.8 (C-Izopr), 94.7 (C-1), 83.9 (C-5′), 78.5 (C-3′), 75.1 (C-4), 74.7 (C-4′), 73.7 (C-2), 71.9 (C-3), 65.4 (C-5), 63.7 (C-1′), 63.2, 62.9 (C-6, C-6′), 29.4 (CH_3_), 19.3 (CH_3_). HRMS (ESI): *m/z* calcd. for C_24_H_32_O_13_Na ([M + Na]^+^) 551.17351; found 551.17349.4′-*O*-Coumaroyl-4,6-*O*-isopropylidene sucrose (**11a**). White foam, ${\left[\alpha \right]}_{D}^{20}$ = +9.8° (*c* 1.0, CH_3_OH). ^1^H NMR (400 MHz, CD_3_OD) δ 7.68 (d, *J* = 15.9 Hz, 1H, H-A), 7.48 (d, *J* = 8.7 Hz, 2H, H-Ph), 6.82 (d, *J* = 8.7 Hz, 2H, H-Ph), 6.38 (d, *J* = 15.9 Hz, 1H, H-B), 5.43 (d, *J* = 4.0 Hz, 1H, H-1), 5.38 (t, *J* = 7.6 Hz, 1H, H-4′), 4.47 (d, *J* = 7.9 Hz, 1H, H-3′), 4.02–3.96 (m, 2H, H-5′, H-5), 3.93 (dd, *J* = 10.2, 5.2 Hz, 1H, H-6a), 3.81 (t, *J* = 9.3 Hz, 1H, H-3), 3.77–3.71 (m, 3H, H-6′a, H-6′b, H-6b), 3.68 (d, *J* = 12.4 Hz, 1H, H-1′a), 3.61 (d, *J* = 12.4 Hz, 1H, H-1′b), 3.53 (dd, *J* = 9.4, 4.1 Hz, 1H, H-2), 3.52 (t, *J* = 9.4 Hz, 1H, H-4), 1.51 (s, 3H, CH_3_) 1.40 (s, 3H, CH_3_). ^13^C NMR (101 MHz, CD_3_OD) δ 168.6 (COO), 161.5 (C-Ph), 147.5 (CH-A), 131.3 (CH-Ph), 127.0 (C-Ph), 116.9 (CH-Ph), 114.5 (CH-B), 105.9 (C-2′), 100.8 (C-Izopr), 94.3 (C-1), 82.5 (C-5′), 77.8 (C-4′), 77.0 (C-3′), 75.1 (C-4), 73.7 (C-2), 72.0 (C-3), 65.4 (C-5), 63.6 (C-1′), 63.5 (C-6′), 63.2 (C-6), 29.5 (CH_3_), 19.4 (CH_3_). HRMS (ESI): *m/z* calcd. for C_24_H_32_O_13_Na ([M + Na]^+^) 551.17351; found 551.17356.6′-*O*-Coumaroyl-4,6-*O*-isopropylidene sucrose (**12a**). White amorphous solid, ${\left[\alpha \right]}_{D}^{20}$ = +52.3° (*c* 1.0, CH_3_OH). ^1^H NMR (400 MHz, CD_3_OD) δ 7.64 (d, *J* = 16.0 Hz, 1H, H-A), 7.46 (d, *J* = 8.7 Hz, 2H, H-Ph), 6.81 (d, *J* = 8.6 Hz, 2H, H-Ph), 6.35 (d, *J* = 16.0 Hz, 1H, H-B), 5.37 (d, *J* = 4.0 Hz, 1H, H-1), 4.44–4.32 (m, 2H, H-6′a, H-6′b), 4.14 (d, *J* = 7.9 Hz, 1H, H-3′), 4.05 (t, *J* = 7.9 Hz, 1H, H-4′), 4.00 (ddd, *J* = 7.9, 6.0, 4.3 Hz, 1H, H-5′), 3.96–3.89 (m, 2H, H-5, H-6a), 3.78 (t, *J* = 9.3 Hz, 1H, H-3), 3.75–3.69 (m, 1H, H-6b), 3.65 (d, *J* = 12.2 Hz, 1H, H-1′a), 3.62 (d, *J* = 12.1 Hz, 1H, H-1′b), 3.49 (t, *J* = 9.4 Hz, 1H, H-4), 3.48 (dd, *J* = 9.4, 4.0 Hz, 1H, H-2), 1.49 (s, 3H, CH_3_), 1.37 (s, 3H, CH_3_). ^13^C NMR (101 MHz, CD_3_OD) δ 169.0 (COO), 161.3 (C-Ph), 146.8 (CH-A), 131.2 (CH-Ph), 127.2 (C-Ph), 116.8 (CH-Ph), 115.0 (CH-B), 105.7 (C-2′), 100.8 (C-Izopr), 94.1 (C-1), 80.8 (C-5′), 78.6 (C-2′), 76.8 (C-4′), 75.3 (C-4), 74.1 (C-2), 72.1 (C-3), 66.5 (C-6′), 65.3 (C-5), 63.5, 63.4 (C-1′, C-6), 29.5 (CH_3_), 19.3 (CH_3_). HRMS (ESI): *m/z* calcd. for C_24_H_32_O_13_Na ([M + Na]^+^) 551.17351; found 551.17339.1′,4′-Di-*O*-coumaroyl-4,6-*O*-isopropylidene sucrose (**13a**). White foam, ${\left[\alpha \right]}_{D}^{20}$ = +30.4° (*c* 1.0, CH_3_OH). ^1^H NMR (400 MHz, CD_3_OD) δ 7.69 (d, *J* = 15.9 Hz, 1H, H-A1), 7.68 (d, *J* = 15.9 Hz, 1H, H-A2), 7.49 (d, *J* = 8.5 Hz, 2H, H-Ph1), 7.47 (d, *J* = 8.5 Hz, 2H, H-Ph2), 6.81 (d, *J* = 8.6 Hz, 2H, H-Ph1), 6.81 (d, *J* = 8.6 Hz, 2H, H-Ph2), 6.39 (d, *J* = 15.8 Hz, 1H, H-B1), 6.38 (d, *J* = 15.9 Hz, 1H, H-B2), 5.50 (d, *J* = 4.0 Hz, 1H, H-1), 5.41 (t, *J* = 7.8 Hz, 1H, H-4′), 4.48 (d, *J* = 8.0 Hz, 1H, H-3′), 4.47 (d, *J* = 12.2 Hz, 1H, H-1′a), 4.32 (d, *J* = 12.2 Hz, 1H, H-1′b), 4.05–3.98 (m, 2H, H-5, H-5′), 3.94 (dd, *J* = 10.3, 5.2 Hz, 1H, H-6a), 3.80 (t, *J* = 9.4 Hz, 1H, H-3), 3.81–3.72 (m, 3H, H-6′a, H-6′b, H-6b), 3.55 (dd, *J* = 9.5, 4.1 Hz, 1H, H-2), 3.54 (t, *J* = 9.4 Hz, 1H, H-4), 1.52 (s, 3H, CH_3_), 1.40 (s, 3H, CH_3_). ^13^C NMR (101 MHz, CD_3_OD) δ 168.5 (COO), 168.4 (COO), 161.5 (C-Ph), 161.4 (C-Ph), 147.6 (CH-A), 147.3 (CH-A), 131.4 (2×CH-Ph), 131.3 (2×CH-Ph), 127.1 (C-Ph), 127.0 (C-Ph), 116.9 (4×CH-Ph), 114.6 (CH-B), 114.4 (CH-B), 104.8 (C-2′), 100.8 (C-Izopr), 94.9 (C-1), 82.6 (C-5′), 77.2 (C-4′), 77.0 (C-3′), 75.2 (C-4), 73.8 (C-2), 72.1 (C-3), 65.5 (C-5), 63.7, 63.6 (C-1′, C-6′), 63.2 (C-6), 29.5 (CH_3_), 19.4 (CH_3_). HRMS (ESI): *m/z* calcd. for C_33_H_38_O_15_Na ([M + Na]^+^) 697.21029; found 697.21004.1′,6′-Di-*O*-coumaroyl-4,6-*O*-isopropylidene sucrose (**14a**). White amorphous solid, ${\left[\alpha \right]}_{D}^{20}$ = −14.5° (*c* 1.0, CH_3_OH). ^1^H NMR (400 MHz, CD_3_OD) δ 7.67 (d, *J* = 15.9 Hz, 1H, H-A1), 7.64 (d, *J* = 16.0 Hz, 1H, H-A2), 7.48 (d, *J* = 8.5 Hz, 2H, H-Ph1), 7.46 (d, *J* = 8.6 Hz, 2H, H-Ph2), 6.81 (d, *J* = 8.8 Hz, 2H, H-Ph1), 6.80 (d, *J* = 8.8 Hz, 2H, H-Ph2), 6.38 (d, *J* = 15.9 Hz, 1H, H-B1), 6.38 (d, *J* = 16.0 Hz, 1H, H-B2), 5.43 (d, *J* = 4.0 Hz, 1H, H-1), 4.46–4.36 (m, 3H, H-1′a, H-6′a, H-6′b), 4.31 (d, *J* = 12.1 Hz, 1H, H-1′b), 4.17 (d, *J* = 8.2 Hz, 1H, H-3′), 4.09 (t, *J* = 8.2 Hz, 1H, H-4′), 4.01 (ddd, *J* = 8.1, 5.8, 4.2 Hz, 1H, H-5′), 3.98–3.93 (m, 1H, H-5), 3.91 (dd, *J* = 10.3, 5.2 Hz, 1H, H-6a), 3.78 (t, 9.2, 1H, H-3), 3.73 (t, *J* = 10.2 Hz, 1H, H-6b), 3.51 (t, *J* = 9.4 Hz, 1H, H-4), 3.50 (dd, *J* = 9.4, 4.2 Hz, 1H, H-2), 1.49 (s, 3H, CH_3_), 1.37 (s, 3H, CH_3_). ^13^C NMR (101 MHz, CD_3_OD) δ 169.0 (COO), 168.4 (COO), 161.4 (C-Ph1), 161.3 (C-Ph2), 147.2 (CH-A1), 146.8 (CH-A2), 131.3 (CH-Ph1), 131.3 (CH-Ph2), 127.2 (C-Ph1), 127.1 (C-Ph2), 116.8 (CH-Ph1), 116.8 (CH-Ph2), 115.0 (CH-B1), 114.7 (CH-B2), 104.6 (C-2′), 100.8 (C-Izopr), 94.6 (C-1), 80.9 (C-5′), 78.6 (C-3′), 76.1 (C-4′), 75.3 (C-4), 74.0 (C-2), 72.1 (C-3), 66.1 (C-6′), 65.4 (C-5), 63.8 (C-1′), 63.3 (C-6), 29.5 (CH_3_), 19.4 (CH_3_). HRMS (ESI): *m/z* calcd. for C_33_H_38_O_15_Na ([M + Na]^+^) 697.21029; found 697.21069.1′-*O*-Feruloyl-4,6-*O*-isopropylidene sucrose (**10b**). White amorphous solid, ${\left[\alpha \right]}_{D}^{20}$ = +14.7° (*c* 1.0, MeOH). ^1^H NMR (400 MHz, CD_3_OD) δ 7.65 (d, *J* = 15.9 Hz, 1H, H-A), 7.19 (d, *J* = 1.9 Hz, 1H, H-Ph), 7.08 (dd, *J* = 8.1, 1.9 Hz, 1H, H-Ph), 6.81 (d, *J* = 8.2 Hz, 1H, H-Ph), 6.39 (d, *J* = 15.9 Hz, 1H, H-B), 5.43 (d, *J* = 4.0 Hz, 1H, H-1), 4.42 (d, *J* = 12.2 Hz, 1H, H-1′a), 4.24 (d, *J* = 12.1 Hz, 1H, H-1′b), 4.17 (d, *J* = 8,6 Hz, 1H, H-3′), 4.09 (t, *J* = 8.5 Hz, 1H, H-4′), 3,93–3,83 (m, 2H, H-5, H-6a), 3.89 (s, 3H, OCH_3_), 3.82–3.67 (m, 4H, H-5′, H-6′a, H-6′b, H-6b), 3.78 (t, *J* = 9.8 Hz, 1H, H-3), 3.52 (dd, *J* = 9.6, 4.0 Hz, 1H, H-2), 3.51 (t, *J* = 9.3 Hz, 1H, H-4), 1.50 (s, 3H, CH_3_), 1.38 (s, 3H, CH_3_). ^13^C NMR (101 MHz, CD_3_OD) δ 168.4 (COO), 150.7 (C-Ph), 149.4 (C-Ph), 147.4 (C-A), 127.6 (C-Ph), 124.3 (CH-Ph), 116.5 (CH-Ph), 114.9 (CH-B), 111.7 (CH-Ph), 104.2 (C-2′), 100.8 (C-Izopr), 94.7 (C-1), 83.9 (C-5′), 78.5 (C-3′), 75.1 (C-4), 74.7 (C-4′), 73.7 (C-2), 71.9 (C-3), 65.4 (C-5), 63.7 (C-1′), 63.2 (C-6), 62.9 (C-6′), 56.5 (OCH_3_), 29.4 (CH_3_), 19.3 (CH_3_). HRMS (ESI): *m/z* calcd. for C_25_H_34_O_14_ [M + Na]^+^ = 581.18408, found 581.18411.4′-*O*-Feruloyl-4,6-*O*-isopropylidene sucrose (**11b**). White foam, ${\left[\alpha \right]}_{D}^{20}$ = +8.2° (*c* 1.0, CH_3_OH). ^1^H NMR (400 MHz, CD_3_OD) δ 7.69 (d, *J* = 15.9 Hz, 1H, H-A), 7.22 (d, *J* = 2.0 Hz, 1H, H-Ph), 7.10 (dd, *J* = 8.2, 2.0 Hz, 1H, H-Ph), 6.82 (d, *J* = 8.1 Hz, 1H, H-Ph), 6.42 (d, *J* = 15.8 Hz, 1H, H-B), 5.43 (d, *J* = 4.0 Hz, 1H, H-1), 5.38 (t, *J* = 7.6 Hz, 1H, H-4′), 4.47 (d, *J* = 7.9 Hz, 1H, H-3′), 4.03–3.96 (m, 2H, H-5′, H-5), 3.94 (dd, *J* = 10.6, 5.2 Hz, 1H, H-6a), 3.90 (s, 3H, OCH_3_), 3.81 (t, *J* = 9.3 Hz, 1H, H-3), 3.77–3.70 (m, 3H, H-6′a, H-6′b, H-6), 3.68 (d, *J* = 12.4 Hz, 1H, H-1′a), 3.61 (d, *J* = 12.3 Hz, 1H, H-1′b), 3.53 (dd, *J* = 9.4, 4.1 Hz, 1H, H-2), 3.52 (t, *J* = 9.4 Hz, 1H, H-4), 1.52 (s, 3H), 1.40 (s, 3H). ^13^C NMR (101 MHz, CD_3_OD) δ 168.6 (COO), 150.9 (C-Ph), 149.4 (C-Ph), 147.7 (CH-A), 127.6 (C-Ph), 124.3 (CH-Ph), 116.5 (CH-Ph), 114.8 (CH-B), 111.8 (CH-Ph), 105.9 (C-2′), 100.8 (C-Izopr), 94.3 (C-1), 82.5 (C-5′), 77.8 (C-4′), 77.0 (C-3′), 75.2 (C-4), 73.9 (C-2), 72.0 (C-3), 65.5 (C-5), 63.6, 63.5 (C-6′, C-1′), 63.3 (C-6), 56.5 (OCH_3_), 29.5 (CH_3_), 19.4 (CH_3_). HRMS (ESI): *m/z* calcd. for C_25_H_34_O_14_ [M + Na]+ = 581.18408; found 581.18409.6′-*O*-Feruloyl-4,6-*O*-isopropylidene sucrose (**12b**). White foam, ${\left[\alpha \right]}_{D}^{20}$ = +49.0° (*c* 1.0, CH_3_OH). ^1^H NMR (400 MHz, CD_3_OD) δ 7.63 (d, *J* = 15.9 Hz, 1H, H-A), 7.20 (d, *J* = 2.0 Hz, 1H, H-Ph), 7.07 (dd, *J* = 8.2, 1.9 Hz, 1H, H-Ph), 6.81 (d, *J* = 8.2 Hz, 1H, H-Ph), 6.40 (d, *J* = 15.9 Hz, 1H, H-B), 5.36 (d, *J* = 4.0 Hz, 1H, H-1), 4.39 (dd, *J* = 11.8, 5.9 Hz, 1H, H-6′a), 4.39 (dd, *J* = 11.5, 4.3 Hz, 1H, H-6′b), 4.15 (d, *J* = 7.9 Hz, 1H, H-3′), 4.05 (t, *J* = 7.9 Hz, 1H, H-4′), 4.00 (ddd, *J* = 7.7, 5.6, 4.2 Hz, 1H, H-5′), 3.96 (bdd, *J* = 10.0, 4.9 Hz, 1H, H-5), 3.91 (dd, *J* = 10.4, 4.4 Hz, 1H, H-6a), 3.90 (s, 3H, OCH_3_), 3.79 (t, *J* = 9.3 Hz, 1H, H-3), 3.72 (t, *J* = 10.1 Hz, 1H, H-6b), 3.67 (d, *J* = 12.4 Hz, 1H, H-1′a), 3.63 (d, *J* = 12.3 Hz, 1H, H-1′b), 3.49 (t, *J* = 9.0 Hz, 1H, H-4), 3.49 (dd, *J* = 9.2, 4.5 Hz, 1H, H-2), 1.49 (s, 3H, CH_3_), 1.37 (s, 3H, CH_3_). ^13^C NMR (101 MHz, CD_3_OD) δ 169.0 (COO), 150.6 (C-Ph), 149.4 (C-Ph), 147.1 (C-A), 127.7 (C-Ph), 124.2 (CH-Ph), 116.5 (CH-Ph), 115.3 (CH-B), 111.7 (CH-Ph), 105.7 (C-2′), 100.8 (C-Izopr), 94.2 (C-1), 80.8 (C-5′), 78.6 (C-3′), 76.8 (C-4′), 75.3 (C-4), 74.1 (C-2), 72.1 (C-3), 66.4 (C-6′), 65.3 (C-5), 2×63.4 (C-1′, C-6), 56.5 (OCH_3_), 29.5 (CH_3_), 19.4 (CH_3_). HRMS (ESI): *m/z* calcd. for C_25_H_34_O_14_ [M + Na]+ = 581.18408; found 581.18412.1′,4′-Di-*O*-feruloyl-4,6-*O*-isopropylidene sucrose (**13b**). Colorless foam, ${\left[\alpha \right]}_{D}^{20}$ = +39.7° (*c* 1.0, CH_3_OH). ^1^H NMR (400 MHz, CD_3_OD) δ 7.69 (d, *J* = 16.0 Hz, 1H, H-A1), 7.67 (d, *J* = 16.0 Hz, 1H, H-A2), 7.21 (d, *J* = 2.2 Hz, 1H, H-Ph1), 7.19 (d, *J* = 1.8 Hz, 1H, H-Ph1), 7.09 (dd, *J* = 8.2, 2.1 Hz, 1H, H-Ph1), 7.08 (dd, *J* = 8.3, 1.9 Hz, 1H, H-Ph2), 6.81 (d, *J* = 8.1 Hz, 2H, H-Ph1, H-Ph2), 6.43 (d, *J* = 16.0 Hz, 1H, H-B1), 6.41 (d, *J* = 15.9 Hz, 1H, H-B2), 5.50 (d, *J* = 4.0 Hz, 1H, H-1), 5.41 (t, *J* = 7.6 Hz, 1H, H-4′), 4.49 (d, *J* = 8.0 Hz, 1H, H-3′), 4.47 (d, *J* = 12.3 Hz, 1H, H-1′a), 4.33 (d, *J* = 12.2 Hz, 1H, H-1′b), 4.05 (ddd, *J* = 10.2, 5.2, 3.5 Hz, 1H, H-5′), 4.05–3.99 (m, 1H, H-5), 3.95 (dd, *J* = 10.3, 5.2 Hz, 1H, H-6a), 3.89 (s, 3H, OCH_3_), 3.88 (s, 3H, OCH_3_), 3.80 (t, *J* = 9.2 Hz, 1H, H-3), 3.79–3.71 (m, 3H, H-6′a, H-6′b, H-6b), 3.55 (dd, *J* = 9.5, 4.1 Hz, 1H, H-2), 3.54 (t, *J* = 9.6 Hz, 1H, H-4), 1.52 (s, 3H, CH_3_), 1.40 (s, 3H, CH_3_). ^13^C NMR (101 MHz, CD_3_OD) δ 168.5 (COO), 168.4 (COO), 150.9 (C-Ph1), 150.8 (C-Ph2), 2×149.4 (C-Ph1,C-Ph2), 147.8 (CH-A1), 147.6 (CH-A2), 2×127.6 (C-Ph1, C-Ph2), 124.4 (CH-Ph1), 124.3 (CH-Ph2), 2×116.5 (CH-Ph1, CH-Ph2), 114.9 (CH-B1), 114.7 (CH-B2), 2×111.7 (CH-Ph1, CH-Ph2), 104.8 (C-2′), 100.8 (C-Izopr), 94.9 (C-1), 82.6 (C-5′), 77.3 (C-4′), 77.0 (C-3′), 75.2 (C-4), 73.8 (C-2), 72.1 (C-3), 65.5 (C-5), 63.7 (C-1′), 63.7 (C-6′), 63.3 (C-6), 2×56.5 (2×OCH_3_), 29.5 (CH_3_), 19.4 (CH_3_). HRMS (ESI): *m/z* calcd. for C_35_H_42_O_17_ [M + Na]+ = 757.23142; found 757.23129.1′,6′-Di-*O*-feruloyl-4,6-*O*-isopropylidene sucrose (**14b**). White amorphous solid, ${\left[\alpha \right]}_{D}^{20}$ = −14.0° (*c* 1.0, CH_3_OH). ^1^H NMR (400 MHz, CD_3_OD) δ 7.65 (d, *J* = 15.9 Hz, 1H, H-A1), 7.62 (d, *J* = 15.9 Hz, 1H, H-A2), 7.19 (d, *J* = 2.0 Hz, 1H, H-Ph1), 7.18 (d, *J* = 2.0 Hz, 1H, H-Ph2), 7.07 (dd, *J* = 8.3, 2.0 Hz, 1H, H-Ph1), 7.05 (dd, *J* = 8.4, 2.0 Hz, 1H, H-Ph2), 6.81 (d, *J* = 8.2 Hz, 1H, H-Ph1), 6.80 (d, *J* = 8.2 Hz, 1H, H-Ph1), 6.42 (d, *J* = 15.9 Hz, 1H, H-B1), 6.40 (d, *J* = 15.9 Hz, 1H, H-B2), 5.42 (d, *J* = 4.0 Hz, 1H, H-1), 4.44 (d, *J* = 12.2 Hz, 1H, H-1′a), 4.42–4.41 (m, 2H, H-6′a, H-6′b), 4.32 (d, *J* = 12.1 Hz, 1H, H-1′b), 4.19 (d, *J* = 8.2 Hz, 1H, H-3′), 4.09 (t, *J* = 8.2 Hz, 1H, H-4′), 4.03 (ddd, *J* = 10.1, 6.9, 4.6 Hz, 1H, H-5′), 3.99 (td, *J* = 10.2, 5.1 Hz, 1H, H-5), 3.91 (dd, *J* = 10.4, 5.2 Hz, 1H, H-6a), 3.88 (s, 6H, OCH_3_), 3.79 (t, *J* = 9.4 Hz, 1H, H-3), 3.73 (t, *J* = 10.5 Hz, 1H, H-6b), 3.51 (t, *J* = 9.5 Hz, 1H, H-4), 3.50 (dd, *J* = 9.5, 4.0 Hz, 1H, H-2), 1.49 (s, 3H, CH_3_), 1.36 (s, 3H, CH_3_). ^13^C NMR (101 MHz, CD_3_OD) δ 169.0 (COO), 168.4 (COO), 150.7 (C-Ph1), 150.6 (C-Ph2), 2×149.3 (C-Ph1,2), 147.5 (CH-A1), 147.1 (CH-A2), 127.7 (C-Ph1), 127.6 (C-Ph2), 2×124.3 (2×CH-Ph1,2), 116.5 (CH-Ph1), 116.4 (CH-Ph2), 115.3 (CH-B1), 114.9 (CH-B2), 2×111.7 (2×CH-Ph1,2), 104.6 (C-2′), 100.8 (C-Izopr), 94.8 (C-1), 80.9 (C-5′), 78.6 (C-3′), 76.1 (C-4′), 75.3 (C-4), 74.0 (C-2), 72.1 (C-3), 66.1 (C-6′), 65.4 (C-5), 63.7 (C-1′), 63.3 (C-6), 2×56.5 (2×OCH_3_), 29.4 (CH_3_), 19.4 (CH_3_). HRMS (ESI): *m/z* calcd. for C_35_H_42_O_17_ [M + Na]+ = 757.23142; found 757.23112

### 3.5. Preparation of 3′-O-acylated Sucrose ***4a*** and ***4b***

Sucrose (**1**) (2.567 g, 7.5 mmol) was suspended in dry DMF (150 mL) and the mixture was stirred at 60 °C until all crystalline sucrose was dissolved. The reaction mixture was then left to cool down to laboratory temperature and triethylamine (8.36 mL, 60 mmol, 8.0 equiv.) and anhydrous CoCl_2_ (0.487 g, 3.75 mmol, 0.5 equiv.) were added. The resulting mixture was stirred for 30 min at laboratory temperature before 4-*O*-acetoxycinnamoyl chloride (2.19 g, 9.75 mmol, 1.3 equiv.) or 3,4,5-trimethoxycinnamoyl chloride (2.503 g, 9.75 mmol, 1.3 equiv.) was added. The dark blue mixture was stirred at laboratory temperature for 2 h. The reaction was stopped by adding acetic acid (4 mL) and the entire reaction mixture was evaporated to dryness. The residue was purified by chromatography on Diaion^®^ HP-20 (water → 30% EtOH/water) to remove cobalt salts, followed by flash chromatography on silica gel eluted with CHCl_3_/MeOH (10:1 → 7:1) affording pure 3′-*O*-(4-*O*-acetoxycinnamoyl) sucrose (1.43 g, 36%) or 3′-*O*-(3,4,5-trimethoxycinnamoyl) sucrose (**4b**) (2.33 g, 55%). 3′-*O*-(4-*O*-Acetoxycinnamoyl) sucrose (1.07 g, 2 mmol) was dissolved in the solution of water and MeOH (1:4, 20 mL) followed by the addition of NH_4_OAc (1.24 g, 16.0 mmol, 8 equiv.). The reaction mixture was stirred at room temperature for 24 h. The reaction was then concentrated and the product was purified by silica gel chromatography (CHCl_3_/MeOH—3:1) to give **4a** (0.87 g, 89%) as a pale yellow amorphous solid.3′-*O*-Coumaroyl sucrose (**4a**). White amorphous solid, ${\left[\alpha \right]}_{D}^{20}$ = −8.8° (*c* 1.0, CH_3_OH), ${\left[\alpha \right]}_{D}^{25}$= −11.0° (*c* 1.03, CH_3_OH) [56]. ^1^H NMR (400 MHz, CD_3_OD) δ 7.71 (d, *J* = 15.9 Hz, 1H, H-A), 7.51 (d, *J* = 8.7 Hz, 2H, H-Ph), 6.81 (d, *J* = 8.7 Hz, 2H, H-Ph), 6.40 (d, *J* = 15.9 Hz, 1H, H-B), 5.46 (d, *J* = 7.9 Hz, 1H, H-3′), 5.43 (d, *J* = 3.7 Hz, 1H, H-1), 4.37 (t, *J* = 7.9 Hz, 1H, H-4′), 3.96–3.88 (m, 2H, H-5′, H-5), 3.85 (dd, *J* = 11.8, 2.3 Hz, 1H, H-6a), 3.87–3.78 (m, 2H, H-6′a, H-6′b), 3.77 (dd, *J* = 12.0, 4.3 Hz, 1H, H-6b), 3.66 (t, *J* = 9.4 Hz, 1H, H-3), 3.66 (d, *J* = 12.2 Hz, 1H, H-1′a), 3.59 (d, *J* = 12.2 Hz, 1H, H-1′b), 3.43 (dd, *J* = 9.8, 3.7 Hz, 1H, H-2), 3.40 (dd, *J* = 9.9, 8.9 Hz, 1H, H-4). ^13^C NMR (101 MHz, CD_3_OD) δ 168.4 (COO), 161.4 (C-Ph), 147.4 (CH-A), 131.4 (2×CH-Ph), 127.2 (C-Ph), 116.8 (2×CH-Ph), 114.8 (CH-B), 104.8 (C-2′), 93.3 (C-1), 84.1 (C-5′), 79.7 (C-3′), 74.9 (C-3), 74.6 (C-5), 73.8 (C-4′), 73.1 (C-2), 71.2 (C-4), 65.3 (C-1′), 62.9 (C-6′), 62.3 (C-6).3′-*O*-(3,4,5-Trimethoxycinnamoyl) sucrose (**4b**). White crystals, m.p.  =  80–83 C; ${\left[\alpha \right]}_{D}^{20}$ = +16.2° (*c* 1.0, CH_3_OH), ${\left[\alpha \right]}_{D}^{25}$ = +5.7° (*c* 1.39, CH_3_OH) [57]; ^1^H NMR (400 MHz, CD_3_OD) δ 7.72 (d, *J* = 15.9 Hz, 1H, H-A), 6.98 (s, 2H, H-Ph), 6.55 (d, *J* = 15.9 Hz, 1H, H-B), 5.48 (d, *J* = 7.8 Hz, 1H, H-3′), 5.44 (d, *J* = 3.7 Hz, 1H, H-1), 4.39 (t*, J* = 7.8 Hz, 1H, H-4), 3.96–3.91 (m, 2H, H-5′, H-5), 3.88 (s, 6H, *m*-OCH_3_), 3.88–3.80 (m, 3H, H-6a, H-6′a, H-6′b), 3.79 (s, 3H, *p*-OCH_3_), 3.79–3.75 (m, 1H, H-6b), 3.67 (dd, *J* = 9.7, 8.9 Hz, 1H, H-3), 3.66 (d, *J* = 12.2 Hz, 1H, H-1′a), 3.58 (d, *J* = 12.2 Hz, 1H, H-1′b), 3.44 (dd, *J* = 9.7, 3.7 Hz, 1H, H-2), 3.40 (dd, *J* = 10.0, 8.9 Hz, 1H, H-4). ^13^C NMR (101 MHz, CD_3_OD) δ 167.7 (COO), 154.8 (2×C-Ph), 147.2 (CH-A), 141.4 (C-Ph), 131.5 (C-Ph), 117.9 (CH-B), 107.0 (2×CH-Ph), 104.8 (C-2′), 93.3 (C-1), 84.2 (C-5′), 79.8 (C-3′), 75.0 (C-3), 74.6 (C-5), 73.9 (C-4), 73.1 (C-2), 71.2 (C-4), 65.4 (C-1′), 62.9 (C-6′), 62.4 (C-6), 61.2 (*p*-OCH_3_), 56.8 (2×*m*-OCH_3_). HRMS (ESI): *m/z* calcd. for C_24_H_34_O_15_ [M + Na]+ = 585.17899; found 585.18030.

### 3.6. General Procedure for the Enzymatic Hydroxycinnamoylation of 3′-O-acylated Sucrose ***4a*** and ***4b***

3′-*O*-Acylated sucrose **4a** or **4b** (1 mmol) was suspended in the appropriate solvent (20 mL), then solid Lipozyme TL IM or Pentopan 500 BG (1 g), molecular sieves 4 Å (1 g) and finally vinyl ester **5a** or **5b** or **5c** (3 mmol, 3 equiv.) were added. The reaction was shaken at 37 °C for the appropriate time (Table 3). The enzyme and sieves were then filtered through Celite 545, the filter cake was washed several times with acetone and the filtrate combined with the washings was concentrated in a rotary evaporator. The residue was purified by column chromatography eluting with EtOAc/MeOH (1:0 → 6:1). The reaction times and yields are summarized in Table 3.6,3′-Di-*O*-coumaroyl sucrose (**15a**). White amorphous solid, ${\left[\alpha \right]}_{D}^{20}$ = −35.6° (*c* 1.0, CH_3_OH), ${\left[\alpha \right]}_{D}^{28}$= −72.0° (*c* 0.3, CH_3_OH) [58]. ^1^H NMR (400 MHz, CD_3_OD) δ 7.71 (d, *J* = 15.9 Hz, 1H, H-A1), 7.64 (d, *J* = 15.9 Hz, 1H, H-A1), 7.51 (d, *J* = 8.7 Hz, 2H, H-Ph1), 7.47 (d, *J* = 8.7 Hz, 2H, H-Ph2), 6.80 (d, *J* = 8.7 Hz, 1H, H-Ph1), 6.79 (d, *J* = 8.6 Hz, 1H, H-Ph2), 6.42 (d, *J* = 16.1 Hz, 1H, H-B2), 6.41 (d, *J* = 15.9 Hz, 1H, H-B1), 5.48 (d, *J* = 7.9 Hz, 1H, H-3′), 5.47 (d, *J* = 4.1 Hz, 1H, H-1), 4.58 (dd, *J* = 11.9, 1.9 Hz, 1H, H-6a), 4.42 (t, *J* = 8.0 Hz, 1H, H-4′), 4.29 (dd, *J* = 11.8, 6.3 Hz, 1H, H-6b), 4.20 (ddd, *J* = 10.1, 6.3, 1.9 Hz, 1H, H-5), 3.97 (ddd, *J* = 7.9, 6.9, 2.9 Hz, 1H, H-5′), 3.91 (dd, *J* = 11.8, 6.9 Hz, 1H, H-6′a), 3.81 (dd, *J* = 11.8, 2.9 Hz, 1H, H-6′b), 3.67 (dd, *J* = 9.7, 8.9 Hz, 1H, H-3), 3.64 (d, *J* = 12.2 Hz, 1H, H-1′a), 3.59 (d, *J* = 12.2 Hz, 1H, H-1′b), 3.47 (dd, *J* = 9.7, 3.8 Hz, 1H, H-2), 3.35 (dd, *J* = 10.0, 8.9 Hz, 1H, H-4). ^13^C NMR (101 MHz, CD_3_OD) δ 169.2 (COO), 168.4 (COO), 161.4 (C-Ph1), 161.3 (C-Ph2), 147.5 (CH-A1), 146.9 (CH-A2), 131.4 (2×CH-Ph1), 131.3 (2×CH-Ph2), 127.2, 127.1 (C-Ph1, C-Ph2), 116.8 (2×CH-Ph1, 2×CH-Ph2), 115.0, 114.7 (CH-B1, CH-B2), 104.8 (C-2′), 92.9 (C-1), 84.2 (C-5′), 79.5 (C-3′), 74.9 (C-3), 74.1 (C-4′), 73.1 (C-2), 72.3 (C-5), 71.8 (C-4), 65.5 (C-1′), 65.2 (C-6), 63.8 (C-6′). HRMS (ESI): *m/z* calcd. for C_30_H_34_O_15_ [M + Na]^+^ = 657.17899, found 657.18914.3′,6′-Di-*O*-coumaroyl sucrose (**16a**). NMR data extracted from the mixture with **15a**, ^1^H NMR (400 MHz, CD_3_OD), δ 7.72 (d, *J* = 15.8 Hz, 1H, H-A1), 7.66 (d, *J* = 15.9 Hz, 1H, H-A2), 7.51 (d, *J* = 8.6 Hz, 1H, 2×H-Ph), 7.46 (d, *J* = 8.8 Hz, 2H, 2×H-Ph), 6.81 (d, *J* = 8.6 Hz, 4H, 4×H-Ph), 6.41 (d, *J* = 15.9 Hz, 1H, H-B1), 6.36 (d, *J* = 16.0 Hz, 1H, H-B2), 5.49 (d, *J* = 8.0 Hz, 1H, H-3′), 5.45 (d, *J* = 3.8 Hz, 1H, H-1), 4.57 (dd, *J* = 11.9, 7.5 Hz, 1H, H-6′a), 4.51 (dd, *J* = 11.9, 3.9 Hz, 1H, H-6′b), 4.45 (t, *J* = 7.6 Hz, 1H, H-4′), 4.19–4.12 (m, 1H, H-5′), 3.94–3.77 (m, 3H, H-5, H-6a, H-6b), 3.68 (t, *J* = 9.5 Hz, 1H, H-3), 3.67 (d, *J* = 12.2 Hz, 1H, H-1′a), 3.61 (d, *J* = 12.0 Hz, 1H, H-1′b), 3.44 (dd, *J* = 9.7, 3.8 Hz, 1H, H-2), 3.41 (t, *J* = 8.9 Hz, 1H, H-4). ^13^C NMR (101 MHz, CD_3_OD) δ 169.0 (COO), 168.3 (COO), 161.3 (C-Ph), 161.2 (C-Ph), 147.5 (CH-A1), 146.9 (CH-A2), 131.4 (2×CH-Ph), 131.2 (2×CH-Ph), 127.5, 127.1 (C-Ph1, C-Ph2), 116.8 (2×CH-Ph1, 2×CH-Ph2), 114.8, 114.6 (CH-B1, CH-B2), 105.0 (C-2′), 93.1 (C-1), 81.2 (C-5′), 79.3 (C-3′), 74.9 (C-3), 74.1 (C-4′), 73.1 (C-2), 72.3 (C-5), 71.8 (C-4), 66.3 (C-6′), 65.1 (C-1′), 62.6 (C-6).6,3′,4′-Tri-*O*-coumaroyl sucrose (**17a**). White foam, ${\left[\alpha \right]}_{D}^{20}$ = −128.2° (*c* 1.0, CH_3_OH). ^1^H NMR (400 MHz, CD_3_OD) δ 7.71 (d, *J* = 15.9 Hz, 1H, H-A1), 7.62 (d, *J* = 15.9 Hz, 1H, H-A2), 7.61 (d, *J* = 16.0 Hz, 1H, H-A3), 7.51 (d, *J* = 8.7 Hz, 2H, H-Ph1), 7.43 (d, *J* = 8.7 Hz, 4H, H-Ph2, H-Ph3), 6.80 (d, *J* = 8.6 Hz, 2H, H-Ph2), 6.79 (d, *J* = 8.7 Hz, 2H, H-Ph1), 6.70 (d, *J* = 8.7 Hz, 2H, H-Ph3), 6.48 (d, *J* = 16.0 Hz, 1H, H-B3), 6.40 (d, *J* = 15.9 Hz, 1H, H-B1), 6.31 (d, *J* = 16.0 Hz, 1H, H-B2), 5.81 (d, *J* = 7.5 Hz, 1H, H-3′), 5.68 (t, *J* = 7.4 Hz, 1H, H-4′), 5.52 (d, *J* = 3.7 Hz, 1H, H-1), 4.67–4.64 (m, 1H, H-6a), 4.34–4.22 (m, 2H, H-6b, H-5), 4.20 (ddd, *J* = 7.4, 4.5, 3.0 Hz, 1H, H-5′), 4.01 (dd, *J* = 12.2, 7.5 Hz, 1H, H-6′a), 3.85 (dd, *J* = 12.1, 3.8 Hz, 1H, H-6′b), 3.71 (dd, *J* = 9.8, 8.9 Hz, 1H, H-3), 3.67 (d, *J* = 12.6 Hz, 1H, H-1′a), 3.62 (d, *J* = 12.4 Hz, 1H, H-1′b), 3.50 (dd, *J* = 9.8, 3.7 Hz, 1H, H-2), 3.33 (t, signal overlapped by CD_3_OD, H-4). ^13^C NMR (101 MHz, CD_3_OD) δ 169.4 (COO), 168.2 (COO), 168.0 COO), 161.5 (C-Ph), 161.42 (C-Ph), 161.1 (C-Ph), 2×147.8 (CH-A1, CH-A2), 146.7 (CH-A3), 131.5 (2×CH-Ph1), 131.4, 131.3 (2×CH-Ph2, 2×CH-Ph3), 127.2 (C-Ph), 127.1 (C-Ph), 127.0 (C-Ph), 116.9, 116.8, 116.7 (2×CH-Ph1, 2×CH-Ph2, 2×CH-Ph3), 115.3 (CH-B3), 114.4 (CH-B1), 114.1 (CH-B2), 105.4 (C-2′), 93.1 (C-1), 83.0 (C-3′), 77.2 (C-3′), 76.3 (C-4′), 74.9 (C-3), 73.0 (C-2), 72.7 (C-5), 72.0 (C-4), 65.6 (C-6), 65.0 (C-1′), 64.1 (C-6′).6,3′,6′-Tri-*O*-coumaroyl sucrose (**18a**). ${\left[\alpha \right]}_{D}^{20}$ = −13.0° (*c* 1.0, CH_3_OH). ^1^H NMR (400 MHz, CD_3_OD) δ 7.72 (d, *J* = 16.0 Hz, 1H, H-A1), 7.62 (d, *J* = 15.8 Hz, 1H, H-A2), 7.60 (d, *J* = 16.0 Hz, 1H, H-A3), 7.52 (d, *J* = 8.7 Hz, 2H, H-Ph1), 7.41 (d, *J* = 8.7 Hz, 2H, H-Ph2), 7.38 (d, *J* = 8.7 Hz, 2H, H-Ph3), 6.81 (d, *J* = 8.8 Hz, 2H, H-Ph1), 6.79 (d, *J* = 8.6 Hz, 2H, H-Ph3), 6.75 (d, *J* = 8.7 Hz, 2H, H-Ph2), 6.42 (d, *J* = 15.9 Hz, 1H, H-B1), 6.41 (d, *J* = 15.9 Hz, 1H, H-B2), 6.28 (d, *J* = 15.9 Hz, 1H, H-B3), 5.53 (d, *J* = 4.1 Hz, 1H, H-1), 5.51 (d, *J* = 8.4 Hz, 1H, H-3′), 4.67 (dd, *J* = 11.4, 1.9 Hz, 1H, H-6a), 4.61 (t, *J* = 8.0 Hz, 1H, H-4′), 4.64–4.47 (m, 2H, H-6′a, H-6′b), 4.32–4.23 (m, 2H, H-6b, H-5), 4.16 (ddd, *J* = 8.1, 6.2, 4.3 Hz, 1H, H-5′), 3.67 (t, *J* = 9.4 Hz, 1H, H-3), 3.66 (d, *J* = 12.3 Hz, 1H, H-1′a), 3.60 (d, *J* = 12.2 Hz, 1H, H-1′b), 3.48 (dd, *J* = 9.7, 4.0 Hz, 1H, H-2), 3.32 (t, signal overlapped by CD_3_OD, H-4). ^13^C NMR (101 MHz, CD_3_OD) δ 169.4 (COO), 168.9 (COO), 168.4 (COO), 161.4 (C-Ph), 161.3 (C-Ph), 161.2 (C-Ph), 147.6 (CH-A1), 146.9, 146.8 (CH-A2, CH-A3), 131.5 (2×CH-Ph1) 131.4 (2×CH-Ph3), 131.2 (2×CH-Ph2), 127.2 (C-Ph), 127.1 (2×C-Ph), 116.8 (4×CH-Ph1,2), 116.7 (2×CH-Ph3), 115.0, 114.9, 114.6 (CH-B1, CH-B2, CH-B3), 105.0 (C-2′), 92.6 (C-1), 81.2 (C-5′), 79.1 (C-3′), 75.0 (C-3), 74.7 (C-4′), 73.1 (C-2), 72.3 (C-5), 72.1 (C-4), 65.6 (C-6, C-6′), 65.4 (C-1′). HRMS (ESI): *m/z* calcd. for C_39_H_40_O_17_ [M + Na]^+^ = 803.21577, found 803.21580.6,3′,1′,6′-Tetra-*O*-coumaroyl sucrose (**19a**, vanicoside D). Amorphous white solid, mp 147.6–154.2 °C [59]; ${\left[\alpha \right]}_{D}^{20}$ = +25.0° (*c* 1.0, CH_3_OH). ^1^H NMR (400 MHz, CD_3_OD) δ 7.72 (d, *J* = 15.9 Hz, 1H, A1), 7.66 (d, *J* = 15.9 Hz, 1H, A2), 7.63 (d, *J* = 16.0 Hz, 1H, A3), 7.61 (d, *J* = 16.0 Hz, 1H, A4), 7.49 (d, *J* = 8.8 Hz, 2H, H-Ph), 7.42 (d, *J* = 8.8 Hz, 2H, H-Ph), 7.41 (d, *J* = 8.5 Hz, 2H, H-Ph), 7.39 (d, *J* = 8.5 Hz, 2H, H-Ph), 6.80 (d, *J* = 8.7 Hz, 2H, H-Ph), 6.78 (d, *J* = 8.7 Hz, 2H, H-Ph), 6.76 (d, *J* = 8.6 Hz, 2H, H-Ph), 6.75 (d, *J* = 8.6 Hz, 2H, H-Ph), 6.44 (d, *J* = 16.0 Hz, 2H, H-Ph), 6.41 (d, *J* = 15.9 Hz, 1H, H-B1), 6.35 (d, *J* = 16.1 Hz, 1H, H-B3), 6.31 (d, *J* = 16.2 Hz, 1H, H-B2), 5.63 (d, *J* = 8.6 Hz, 1H, H-B4), 5.63 (d, *J* = 8.6 Hz, 1H, H-3′), 5.58 (d, *J* = 3.8 Hz, 1H, H-1), 4.69 (t, *J* = 8.7 Hz, 1H, H-4′), 4.69 (dd, *J* = 11.3, 1.5 Hz, 1H, H-6a), 4.62–4.46 (m, 2H, H-6′a, H-6′b), 4.35 (d, *J* = 12.0 Hz, 1H, H-1′a), 4.32 (d, *J* = 12.0 Hz, 1H, H-1′b), 4.31–4.24 (m, 1H, H-5), 4.29 (dd, *J* = 11.0, 8.8 Hz, 1H, H-6a), 4.19 (ddd, *J* = 9.3, 5.2, 3.0 Hz, 1H, H-5′), 3.66 (t, *J* = 9.3 Hz, 1H, H-3), 3.47 (dd, *J* = 9.7, 4.0 Hz, 1H, H-2), 3.34 (t, *J* = 9.1 Hz, 1H, H-4). HRMS (ESI): *m/z* calcd. for C_48_H_46_O_19_ [M + Na]^+^ = 949.25255, found 949.25200.6-*O*-feruloyl-3′-*O*-coumaroyl sucrose (**15b**). White foam, ${\left[\alpha \right]}_{D}^{20}$ = −54.9° (*c* 1.0, CH_3_OH). ^1^H NMR (400 MHz, CD_3_OD) δ 7.71 (d, *J* = 15.9 Hz, 1H, H-A1), 7.62 (d, *J* = 15.8 Hz, 1H, H-A2), 7.51 (d, *J* = 8.7 Hz, 2H, H-Cou), 7.22 (d, *J* = 2.2 Hz, 1H, H-Fer), 7.06 (dd, *J* = 8.2, 1.9 Hz, 1H, H-Fer), 6.80 (d, *J* = 8.6 Hz, 2H, H-Cou), 6.79 (d, *J* = 8.4 Hz, 1H, H-Fer), 6.45 (d, *J* = 15.9 Hz, 1H, H-B2), 6.42 (d, *J* = 15.9 Hz, 1H, H-B1), 5.49 (d, *J* = 7.9 Hz, 1H, H-3′), 5.48 (d, *J* = 4.3 Hz, 1H, H-1), 4.60 (dd, *J* = 11.5, 1.4 Hz, 1H, H-6a), 4.46 (t, *J* = 8.1 Hz, 1H, H-4′), 4.27 (dd, *J* = 11.4, 6.8 Hz, 1H, H-6b), 4.27–4.18 (m, 1H, H-5), 3.97 (td, *J* = 7.5, 2.8 Hz, 1H, H-5′), 3.91 (dd, *J* = 11.8, 7.0 Hz, 1H, H-6′a), 3.87 (s, 3H, OCH_3_), 3.82 (dd, *J* = 11.8, 2.9 Hz, 1H, H-6′b), 3.67 (t, *J* = 9.3 Hz, 1H, H-3), 3.64 (d, *J* = 12.2 Hz, 1H, H-1′a), 3.59 (d, *J* = 12.2 Hz, 1H, H-1′b), 3.47 (dd, *J* = 9.7, 3.8 Hz, 1H, H-2), 3.33 (t, *J* = 9.4 Hz, 1H, H-4). ^13^C NMR (101 MHz, CD_3_OD) δ 169.2 (COO), 168.4 (COO), 161.4 (C-Cou), 150.6 (C-Fer), 149.4 (C-Fer), 147.5 (CH-A1), 147.1 (CH-A2), 131.4 (2×CH-Cou), 127.7 (C-Fer), 127.2 (C-Cou), 124.3 (CH-Fer), 116.8 (2×CH-Cou), 116.4 (CH-Fer), 115.4 (CH-B2), 114.7 (CH-B1), 111.6 (CH-Fer), 104.8 (C-2′), 92.8 (H-1), 84.2 (C-5′), 79.4 (C-3′), 75.0 (C-3), 74.2 (C-4′), 73.1 (C-2), 72.3 (C-5), 71.9 (C-4), 65.5 (C-1′), 65.4 (C-6), 63.9 (C-6′), 56.5 (OCH_3_). HRMS (ESI): *m/z* calcd. for C_31_H_36_O_16_ [M + Na]^+^ = 687.18956, found 687.18949.3′-*O*-coumaroyl-6′-*O*-feruloyl sucrose (**16b**). NMR data extracted from the mixture with **15b** (reaction with Pentopan, Table 3, entry 5); ^1^H NMR (400 MHz, CD_3_OD) δ 7.72 (d, *J* = 15.8 Hz, 1H, H-A1), 7.66 (d, *J* = 15.9 Hz, 1H, H-A2), 7.50 (d, *J* = 8.8 Hz, 2H, H-Cou), 7.18 (d, *J* = 2.0 Hz, 1H, H-Fer), 7.09 (dd, *J* = 8.2, 1.9 Hz, 1H, H-Fer), 6.82 (d, *J* = 8.2 Hz, 2H, H-Fer), 6.81 (d, *J* = 8.8 Hz, 1H, H-Cou), 6.41 (d, *J* = 16.0 Hz, 1H, H-B1), 6.40 (d, *J* = 15.9 Hz, 1H, H-B2), 5.49 (d, *J* = 8.0 Hz, 1H, H-3′), 5.45 (d, *J* = 3.7 Hz, 1H, H-1), 4.57 (dd, *J* = 11.7, 7.4 Hz, 1H, H-6′a), 4.51 (dd, *J* = 11.8, 3.7 Hz, H-6′b), 4.45 (t, *J* = 7.9 Hz, 1H, H-4′), 4.18 (td, *J* = 7.7, 3.9 Hz, 1H, H-5′), 3.97 (dt, *J* = 7.7, 3.6 Hz, 1H, H-5), 3.92 (bd, *J* = 10.4 Hz, 1H, H-6a), 3.82–3.78 (m, 1H, H-6b), 3.89 (s, 3H, OCH_3_), 3.68 (t, *J* = 9.4 Hz, 1H, H-3), 3.67 (d, *J* = 12.5 Hz, 1H, H-1′a), 3.61 (d, *J* = 12.4 Hz, 1H, H-1′b), 3.44 (dd, *J* = 9.7, 3.7 Hz, 1H, H-2), 3.41 (t, *J* = 9.4 Hz, 1H, H-4). ^13^C NMR (101 MHz, CD_3_OD) δ 169.0 (COO), 168.3 (COO), 161.3 (C-Cou), 150.6 (C-Fer), 149.3 (C-Fer), 147.5 (CH-A1), 147.2 (CH-A2), 131.4 (2×CH-Cou), 127.7 (C-Fer), 127.1 (C-Cou), 124.2 (CH-Fer), 116.8 (2×CH-Cou), 116.5 (CH-Fer), 115.2 (CH-B2), 114.6 (CH-B1), 111.7 (CH-Fer), 105.1 (C-2′), 93.1 (H-1), 81.2 (C-5′), 79.3 (C-3′), 2×75.0 (C-3, C-4′), 74.4 (C-5), 73.2 (C-2), 71.4 (C-4), 66.3 (C-6′), 65.1 (C-1′), 62.6 (C-6), 56.5 (OCH_3_).6,4′-di-*O*-feruloyl-3′-*O*-coumaroyl sucrose (**17b**). NMR data extracted from the fraction with **15b** (reaction with Lipozyme, Table 3, entry 4). ^1^H NMR (400 MHz, CD_3_OD) δ 7.71 (d, *J* = 15.8 Hz, 1H, H-A), 7.60 (d, *J* = 15.9 Hz, 1H, H-2), 7.56 (d, *J* = 15.8 Hz, 1H, H-3), 7.50 (d, *J* = 8.6 Hz, 2H, H-Cou), 7.13 (d, *J* = 2.0 Hz, 1H, H-Fer1), 7.09 (d, *J* = 2.0 Hz, 1H, H-Fer2), 7.03 (dd, *J* = 8.4, 2.2 Hz, 1H, H-Fer1), 7.01 (dd, *J* = 8.4, 2.2 Hz, 1H, H-Fer2), 6.79 (d, *J* = 8.1 Hz, 1H, H-Fer2), 6.79 (d, *J* = 8.7 Hz, 2H, H-Cou), 6.70 (d, *J* = 8.1 Hz, 1H), 6.49 (d, *J* = 15.4 Hz, 1H, H-B2), 6.41 (d, *J* = 15.8 Hz, 1H, H-B1), 6.29 (d, *J* = 15.9 Hz, 1H, H-B3), 5.82 (d, *J* = 7.6 Hz, 1H, H-3′), 5.68 (t, *J* = 7.5 Hz, 1H, H-4′), 5.53 (d, *J* = 3.6 Hz, 1H, H-1), 4.68 (bd, *J* = 10.9 Hz, 1H, H-6a), 4.35–4.28 (m, 1H, C-5), 4.29–4.24 (m, 1H, H-6b), 4.24–4.17 (m, 1H, H-5′), 4.01 (dd, *J* = 12.1, 7.2 Hz, 1H, H-6′a), 3.90 (dd, *J* = 12.0, 3.7, 1H, H-6′b), 3.87 (s, 3H, OCH_3_), 3.80 (s, 3H, OCH_3_), 3.71 (t, *J* = 9.4 Hz, 1H, H-3), 3.67 (d, *J* = 12.4 Hz, 1H, H-1′a), 3.63 (d, *J* = 12.2 Hz, 1H, H-1′b), 3.51 (dd, *J* = 9.8, 3.8 Hz, 1H, H-2), 3.32 (t, *J* = 9.7 Hz, 1H, H-4). ^13^C NMR (101 MHz, MeOD) δ 169.4 (COO), 168.2 (COO), 168.0 (COO), 161.4 (C-Cou), 150.8 (C-Fer1), 150.5 (C-Fer2), 149.3 (C-Fer1), 149.2 (C-Fer2), 148.0 (CH-A3), 147.9 (CH-A1), 146.9 (C-A2), 131.5 (2×CH-Cou), 127.8 (C-Fer), 127.5 (C-Fer), 127.1 (C-Cou), 124.4 (CH-Fer2), 124.2 (CH-Fer1), 116.8 (2×CH-Cou), 116.5 (CH-Fer2), 116.4 (CH-Fer1), 115.6 (CH-B2), 114.4 (CH-B1), 114.3 (CH-B3), 111.9 (CH-Fer1), 111.8 (CH-Fer2), 105.3 (C-2′), 93.1 (C-1), 82.9 (C-5′), 77.1 (C-3′), 76.4 (C-4′), 74.9 (C-3), 73.0 (C-2), 72.8 (C-5), 72.0 (C-4), 65.7 (C-6), 65.0 (C-1′), 64.1 (C-6′), 56.5 (2×OCH_3_).6,6′-Di-*O*-feruloyl-3′-*O*-coumaroyl sucrose (**18b**). White foam, ${\left[\alpha \right]}_{D}^{20}$ = −15.0° (*c* 1.0, CH_3_OH). ^1^H NMR (400 MHz, CD_3_OD) δ 7.72 (d, *J* = 15.9 Hz, 1H, H-A1)), 7.60 (d, *J* = 15.9 Hz, 1H, H-A2), 7.57 (d, *J* = 16.0 Hz, 1H, H-A3), 7.51 (d, *J* = 8.7 Hz, 2H, H-Cou), 7.19 (d, *J* = 1.9 Hz, 1H, H-Fer1), 7.08 (d, *J* = 1.9 Hz, 1H, H-Fer2), 6.99 (dd, *J* = 8.3, 2.0 Hz, 1H, H-Fer1), 6.98 (dd, *J* = 8.3, 2.0 Hz, 1H, H-Fer2), 6.80 (d, *J* = 9.0 Hz, 2H, H-Cou), 6.78 (d, *J* = 8.7 Hz, 1H, H-Fer2), 6.74 (d, *J* = 8.1 Hz, 1H, H-Fer1), 6.46 (d, *J* = 15.8 Hz, 1H, H-B2), 6.43 (d, *J* = 15.9 Hz, 1H, H-B1), 6.28 (d, *J* = 15.9 Hz, 1H, H-B3), 5.52 (d, *J* = 8.1 Hz, 1H, H-3′), 5.52 (d, *J* = 4.2 Hz, 1H, H-1), 4.70 (dd, *J* = 11.5, 1.5 Hz, 1H, H-6a), 4.64 (t, *J* = 8.2 Hz, 1H, H-4′), 4.60 (dd, *J* = 11.9, 4.1 Hz, 1H, H-6′a), 4.52 (dd, *J* = 11.9, 6.5 Hz, 1H, 6′b), 4.35–4.24 (m, 1H, H-5), 4.22 (dd, *J* = 7.5, 11.6 Hz, 1H, H-6b), 4.17 (ddd, *J* = 8.1, 6.5, 4.1 Hz, 1H, H-5′), 3.87 (s, 3H, OCH_3_), 3.84 (s, 3H, OCH_3_), 3.67 (t, *J* = 9.3 Hz, 1H, H-3), 3.65 (d, *J* = 12.1 Hz, 1H, H-1′a), 3.60 (d, *J* = 12.2 Hz, 1H, H-1′b), 3.47 (dd, *J* = 9.8, 3.8 Hz, 1H, H-2), 3.29 (dd, *J* = 9.9, 9.0 Hz, 1H, H-4). ^13^C NMR (101 MHz, CD_3_OD) δ 169.3 (COO), 168.8 (COO), 168.4 (COO), 161.4 (C-Cou), 150.6 (C-Fer1), 150.6 (C-Fer2), 149.3 (C-Fer1), 149.3 (C-Fer2), 147.6 (CH-A1), 147.2 (CH-A2), 147.0 (CH-A3), 131.5 (2×CH-Cou), 127.7 (C-Fer1), 127.7 (C-Fer2), 127.2 (C-Cou), 124.5 (CH-Fer1), 124.1 (CH-Fer2), 116.8 (2×CH-Cou), 116.5 (CH-Fer2), 116.3 (CH-Fer1), 115.3 (CH-B2), 115.2 (CH-B3), 114.6 (CH-B1), 111.7 (CH-Fer2), 111.6 (CH-Fer2), 104.9 (C-2′), 92.6 (C-1), 81.2 (C-5′), 79.1 (C-3′), 75.0 (C-3), 74.7 (C-4′), 73.1 (C-2), 72.4 (C-5), 72.2 (C-4), 65.8 (C-6), 65.7 (C-6′), 65.5 (C-1′), 56.5 (OCH_3_), 56.4 (OCH_3_). HRMS (ESI): *m/z* calcd. for C_41_H_44_O_19_ [M + Na]^+^ = 863.23690, found 893.23625.6,3′-Di-*O*-(3,4,5-tri-*O*-methoxycinnamoyl) sucrose (**15c**, glomeratose D). White solid, m.p.  =  101–103 °C; ${\left[\alpha \right]}_{D}^{20}$ = −49.7° (c 1.0, MeOH), ${\left[\alpha \right]}_{D}^{28}$ = −55.8° (c 0.53, MeOH) [57]. ^1^H NMR (400 MHz, CD_3_OD) δ 7.69 (d, *J* = 16.0 Hz, 1H, H-A1), 7.61 (d, *J* = 15.9 Hz, 1H, H-A2), 6.95 (s, 2H, H-Ph1), 6.91 (s, 2H, H-Ph2), 6.55 (d, *J* = 16.0 Hz, 1H, H-B2), 6.54 (d, *J* = 15.7 Hz, 1H, H-B1), 5.53 (d, *J* = 8.0 Hz, 1H, H-3′), 5.52 (d, *J* = 3.8 Hz, 1H, H-1), 4.71 (d, *J* = 11.6, 1.6 Hz, 1H, H-6a), 4.51 (t, *J* = 8.0 Hz, 1H, H-4′), 4.32–4.26 (m, 1H, H-5), 4.21 (d, *J* = 11.5, 7.4 Hz 1H, H-6b), 3.98 (ddd, *J* = 8.1, 6.8, 3.2 Hz, 1H, H-5′), 3.87 (s, 6H, 2×*m*-OCH_3_), 3.85 (s, 6H, 2×*m*-OCH_3_), 3.92–3.80 (m, 2H, H-6′a, H-6′b, overlapped by OCH_3_ groups), 3.79 (s, 3H, *p*-OCH_3_), 3.78 (s, 3H, *p*-OCH_3_), 3.67 (dd, *J* = 9.7, 8.8 Hz, 1H, H-3), 3.63 (d, *J* = 12.1 Hz, 1H, H-1′a), 3.58 (d, *J* = 12.2 Hz, 1H, H-1′b), 3.48 (dd, *J* = 9.6, 4.0 Hz, 1H, H-2), 3.31 (t, *J* = 9.6 Hz, 1H, H-4, overlapped by CD_3_OD). ^13^C NMR (101 MHz, CD_3_OD) δ 168.6 (COO), 167.8 (COO), 154.8 (2×C-Ph1), 154.7 (2×C-Ph2), 147.2 (C-A1), 146.6 (C-A2), 141.3 (C-Ph1), 141.2 (C-Ph2), 131.5 (C-Ph1) 131.5 (C-Ph2), 118.1 (CH-B1), 117.8 (CH-B2), 106.9 (2×CH-Ph1), 106.8 (2×CH-Ph2), 104.8 (C-2′), 92.7 (C-1), 84.3 (C-5′), 79.4 (C-3′), 75.1 (C-3), 74.2 (C-4′), 73.1 (C-2), 72.5 (C-5), 72.0 (C-4), 65.8, 65.7 (C-1′, C-6), 63.8 (C-6′), 61.2 (*p*-OCH_3_), 61.1 (*p*-OCH_3_), 56.8 (2×*m*-OCH_3_), 56.7 (2×*m*-OCH_3_). HRMS (ESI): *m/z* calcd. for C_36_H_46_O_19_ [M + Na]+ = 805.25255; found 805.25299.6,3′,4′-Di-*O*-(3,4,5-tri-*O*-methoxycinnamoyl) sucrose (**16c**). Selected signals from impurity below 5% in fraction of **17c**. ^1^H NMR (400 MHz, CD_3_OD) δ 5.86 (d, *J* = 7.5 Hz, 1H), 5.73 (t, *J* = 7.5 Hz, 1H), 5.57 (d, *J* = 3.9 Hz, 1H).6,3′,6′-Tri-*O*-(3,4,5-tri-*O*-methoxycinnamoyl) sucrose (**17c**). White solid; ${\left[\alpha \right]}_{D}^{20}$ = −10.0° (*c* 1.0, CH_3_OH) ^1^H NMR (400 MHz, CD_3_OD) δ 7.70 (d, *J* = 15.9 Hz, 1H, H-A1), 7.57 (d, *J* = 16.1 Hz, 1H, H-A2), 7.53 (d, *J* = 16.1 Hz, 1H, H-A3), 6.95 (s, 2H, H-Ph1), 6.84 (s, 2H, H-Ph2), 6.78 (s, 2H, H-Ph3), 6.55 (d, *J* = 16.0 Hz, 1H, H-B1), 6.54 (d, *J* = 15.9 Hz, 1H, H-B2), 6.37 (d, *J* = 15.9 Hz, 1H, H-B3), 5.56 (d, *J* = 7.8 Hz, 1H, H-3′), 5.53 (d, *J* = 3.9 Hz, 1H, H-1), 4.79 (dd, *J* = 11.8, 1.7 Hz, 1H, H-6a), 4.68 (t, *J* = 7.9 Hz, 1H, H-4′), 4.62 (dd, *J* = 11.9, 4.3 Hz, 1H, H-6′a), 4.53 (dd, *J* = 11.9, 6.5 Hz, 1H, H-6′b), 4.35 (ddd, *J* = 10.1, 8.1, 1.7 Hz, 1H, H-5), 4.20 (ddd, *J* = 10.0, 6.4, 4.2 Hz, 1H, H-5′), 4.17 (dd, *J* = 11.9, 8.1 Hz, 1H, H-6b), 3.86 (s, 6H, 2×*m*-OCH_3_), 3.83 (s, 6H, 2×*m*-OCH_3_), 3.79 (s, 6H, *m*-OCH_3_), 3.78 (s, 3H, *p*-OCH_3_), 3.77 (s, 3H, *p*-OCH_3_), 3.72 (s, 3H, *p*-OCH_3_), 3.68 (dd, *J* = 9.8, 8.8 Hz, 1H, H-3), 3.65 (d, *J* = 12.2 Hz, 1H, H-1′a), 3.60 (d, *J* = 12.3 Hz, 1H, H-1′b), 3.49 (dd, *J* = 9.7, 3.9 Hz, 1H, H-2), 3.27 (dd, *J* = 10.1, 8.8 Hz, 1H, H-4). HRMS (ESI): *m/z* calcd. for C_48_H_58_O_23_ [M + Na]^+^ = 1025.32611, found 1025.32666.2,6-Di-*O*-(3,4,5-tri-*O*-methoxycinnamoyl) sucrose (**20**). White amorphous solid, t.t.  =  171–174 C; ${\left[\alpha \right]}_{D}^{20}$ = + 28.0° (c 0.5, MeOH); ^1^H NMR (400 MHz, CD_3_OD) δ 7.74 (d, *J* = 15.9 Hz, 1H, H-A1), 7.67 (d, *J* = 16.0 Hz, 1H, H-A2), 6.98 (s, 2H, H-Ph1), 6.95 (s, 2H, H-Ph2), 6.58 (d, *J* = 15.9 Hz, 1H, H-B2), 6.54 (d, *J* = 16.0 Hz, 1H, H-B1), 5.62 (d, *J* = 3.8 Hz, 1H, H-1), 4.75 (dd, *J* = 10.1, 3.7 Hz, 1H, H-2), 4.56 (dd, *J* = 11.9, 1.8 Hz, 1H, H-6a), 4.32 (dd, *J* = 11.8, 6.6 Hz, 1H, H-6b), 4.22 (d, *J* = 8.8 Hz, 1H, H-3′), 4.23–4.19 (m, 1H, H-5), 4.08 (t, *J* = 8.7 Hz, 1H, H-4′), 4.02 (dd, *J* = 10.1, 8.9 Hz, 1H, H-3), 3.88 (s, 6H, *m*-OCH_3_), 3.88 (s, 6H, *m*-OCH_3_), 3.87–3.81 (m, 2H, H-6′a, H-5′, overlapped with signals of OCH_3_), 3.80 (s, 3H, *p*-OCH_3_), 3.80 (s, 3H, *p*-OCH_3_), 3.76 (dd, *J* = 12.1, 6.2 Hz, 1H, 6′b), 3.53 (d, *J* = 11.9 Hz, 1H, H-1′a), 3.45 (dd, *J* = 10.1, 8.9 Hz, 1H, H-4), 3.32 (d, *J* = 11.9 Hz, 1H, H-1′b). ^13^C NMR (101 MHz, CD_3_OD) δ 168.6 (COO), 168.3 (COO), 154.9 (2×C-Ph1), 154.8 (2×C-Ph2), 147.2 (C-A1), 146.6 (C-A2), 141.5 (C-Ph1), 141.3 (C-Ph2), 131.6 (C-Ph1) 131.5 (C-Ph2), 118.1 (CH-B2), 117.9 (CH-B1), 106.9 (2×CH-Ph1), 106.8 (2×CH-Ph2), 105.7 (C-2′), 90.5 (C-1), 84.0 (C-5′), 77.1 (C-3′), 75.7 (C-4′), 74.6 (C-2), 72.1 (C-4), 72.0 (C-3), 71.9 (C-5), 65.3 (C-6), 64.2 (C-6′), 63.1 (C-1′), 61.2 (2×*p*-OCH_3_), 56.8 (4x*m*-OCH_3_). HRMS (ESI): *m/z* calcd. for C_36_H_46_O_19_ [M + Na]+ = 805.25255; found 805.25263.

## 4. Conclusions

The first preparative enzymatic acylation of free and derivatized sucrose with substituted cinnamoyl donors was studied. Two commercial immobilized enzyme preparations exhibiting feruloyl esterase activity were tested, namely Lipozyme TL IM and Pentopan 500 BG. When the reactions on all sucrose acceptors are evaluated, acylations catalyzed by Pentopan were much slower and proceeded with higher regioselectivity to primary OH groups. Pentopan had a narrower substrate specificity for donors and was more sensitive to the choice of solvent. It did not react with hydrophobic vinyl trimethoxycinnamate and catalyzed the production of ferulates better than coumarates. It reacted with all acceptors except sucrose, where there was a problem with the solubility and sensitivity of Pentopan to other solvents. The sucrose diisopropylidene was hydroxycinnamoylated by Pentopan to 6′-OH, while this position did not react with Lipozyme. Lipozyme worked well with all cinnamates, and best with coumarate. Reactions of Lipozyme with vinyl coumarate often yielded triacylation products in isolable amounts. In the case of free and isopropylidened sucrose, it preferred the 1′-OH and even 4′-OH position over 6′-OH in acylation. If 3′-*O*-cinnamoyl sucrose was used as an acceptor, the regioselectivity of Lipozyme-catalyzed acylations changed. 6-O-Cinnamates were isolated as major products and minor 6,6′- more than 6,4′-diacylation products were minor. In conclusion, it can be stated that the results of this work show that the regioselectivity of a particular enzymatic acylation of sucrose can be controlled by the use of a suitable enzyme, the modification of the acceptor (sucrose), and also the use of a suitable reaction medium.

It should be mentioned that several natural PPES (e.g., glomeratose D, vanicoside D) were already prepared within this study of acylation methods. This work was carried out with the aim of preparing precursors of biologically active PPES such as vanicoside B. In conclusion, it can be said that its effective preparation by a chemoenzymatic method will not be straightforward. However, with a suitable choice of starting substances, enzymes and acyl donors, enzyme acylation can significantly simplify the preparation of such compounds. Our further studies in this area will also go in this direction.

## Data Availability

Data are contained within the article.

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
