# Peer review of "An Exploratory Study of the Enzymatic Hydroxycinnamoylation of Sucrose and Its Derivatives"

_molecules, 2024, doi:10.3390/molecules29174067_

Round 1

Reviewer 1 Report

Comments and Suggestions for Authors

The authors have conducted a pilot study on the enzymatic catalysis of hydroxycinnamoylation of sucrose and its derivatives. This study evaluated the substrate specificity of two commercial enzymes towards donors and acceptors, as well as the regioselectivity of the reactions. These findings provide significant insights into the understanding and realization of the biosynthesis of phenylpropanoid sucrose esters. As an exploratory study, I recommend that this paper be published; however, several minor issues need to be addressed first:

1. I recommend moving all experimental details from the Results and Discussion section to the Methods section for better organization.

2. As the authors mentioned, the choice of solvent significantly affects the solubility of sucrose and the enzymatic activity. I am curious whether the authors investigated the effects of the solvent on the specificity and regioselectivity of free sucrose.

3. There is no chromatography data provided. Either remove references such as "fifth fraction" or add the necessary details, as readers may not understand what these terms refer to.

4.I suggest annotating the yields next to the structures in schemes for better readability.

Reviewer 2 Report

Comments and Suggestions for Authors

Dear Editor and authors, I recommend the publication of this manuscript after minor revision.

The following changes are recommended and some clarifications should be made:

-        Potential application of the present results could be included at the end of the Abstract section.

-        Any reference for these statements: "Despite the fact that molecular techniques and genome sequencing have developed rapidly and the ABCD classification system no longer adequately represents the evolutionary relationships of fungal and bacterial FAEs, for our chemical reasoning this system works the best. On the basis of the classification in terms of their substrate specificity, type A FAEs show a preference for the phenolic moiety of the substrate that contains methoxy substitutions. These enzymes appear to prefer hydrophobic substrates with bulky substituents on the benzene ring. Their protein sequences are more similar to the lipase sequences. Type B FAEs prefer to hydrolyze substrates with hydroxyl substituents and have sequence homology similar to acetyl xylan esterases. Finally, type C and D FAEs have broader substrate specificities with activities towards all model substrates and share sequence homology with chlorogenic acid esterases and tannases, while type D share sequence homology with xylanases and are the only group capable of hydrolyzing diferulates."

-        The aim of the study should be clearly presented at the end of Introduction section.

-        The authors could include future investigations related to the present study in the section Conclusions.

Reviewer 3 Report

Comments and Suggestions for Authors

Dear editor and authors,

The manuscript entitled "An Exploratory Study of the Enzymatic Hydroxycinnamoylation of Sucrose and its Derivatives" evaluated the potential of the enzymatic hydroxycinnamoylation of sucrose and its derivatives. It presents scientific relevance for Chemistry, Pharmacy and others area. However, you need to change some details/information in the Abstract, Introduction, Material and Methods, results and discussion, and conclusions.

1. Abstract: Adequate! But:

- The abstract is well written, but there is few information about the methods used. Also, I suggest inserting the results obtained (numerical data!!!) more relevant. At the end, I suggest highlighting the advantages/ disadvantages of the study and methods.

2. Introduction section:

- Adequate! At end, I suggest highlighting the objective and advantages/disadvantages/limitations of the study.

3. Results and discussion

- Page 6, lines 248-253: The authors wrote: “…For enzymes that normally hydrolyze amide or ester bonds to work in the esterification direction, the reactions have to be carried out in an organic medium with only the minimum amount of water necessary to maintain their active structure. In such reactions, it is difficult to find a suitable solvent and temperatures that are compatible with the solubility of all substrates in the reaction and the stability and activity of the enzyme.” I suggest expanding the discussions!

- Pages 6-10, in “2.3. Enzymatic hydroxycinnamoylation of 2,1':4,6-di-O-isopropylidene sucrose 2”; “2.3. Enzymatic hydroxycinnamoylation of 4,6-O-isopropylidene sucrose 3” and “2.3. Enzymatic cinnamoylation of 3'-O-acylated sucroses 4a and 4b” sections: 2.3 or 2.4? I suggest expanding the discussions and comparing the results obtained with other works in the literature. The results are interesting!

- I suggest discussing the reaction “yields” further, as well as improving the presentations in Tables 2 and 3.

- I suggest, at the end of the "results and discussion", to write a paragraph summarizing the findings focusing on the advantages/disadvantages/limitations of the study and methods.

4. Materials and methods section: The methodological proposal is appropriate to the manuscript, but I suggest:

- Page 10, in “43.1. General” section: Long paragraph containing a lot of information. I suggest dividing the text into 2 or 3 paragraphs and citing the information in steps!

- Page 10, in “3.2. Feruloylation of Free Sucrose Using Lipozyme TL IM” section: Any previous protocols/studies? If yes, indicate reference! I suggest indicating the "Selected NMR signals of products" as results, in the form of "supplementary material".

- Pages 12-16, in “3.3. Enzymatic hydroxycinnamoylation of 2,1':4,6-di-O-isopropylidene sucrose 2” and “3.4. Enzymatic hydroxycinnamoylation of 4,6-O-isopropylidene sucrose 3” sections: I suggest indicating the "Selected NMR signals of products" as results, in the form of "supplementary material".

- Pages 16-20, in “3.5. Preparation of 3´-O-acylated sucrose 4a and 4b” and “3.6. General procedure for the enzymatic hydroxycinnamoylation of 3´-O-acylated sucrose 4a and 4b” sections: Any previous protocols/studies? If yes, indicate reference! I suggest indicating the "Selected NMR signals of products" as results, in the form of "supplementary material".

5. Conclusion: I suggest inserting the results obtained more relevant. I suggest pointing out the main results and disadvantages/limitations of the method and the study!

6. Table and Figures: Adequate. I suggest discussing the reaction “yields” further, as well as improving the presentations in Tables 2 and 3.

7. References: Please, check if the references are in accordance with the journal's rules.

Comments on the Quality of English Language

The language (English) is satisfactory (but, I suggest the final revision)!
